# Bayesian Sensitivity of Causal Inference Estimators under Evidence-Based Priors

**Nikita Dhawan**                                                              *nikita@cs.toronto.edu*
*University of Toronto*
*Vector Institute*

**Daniel Shen**                                                               *d2shen@uwaterloo.ca*
*University of Waterloo*

**Leonardo Cotta**                                           *leonardo.cotta@vectorinstitute.ai*
*Vector Institute*

**Chris J. Maddison**                                                      *cmaddis@cs.toronto.edu*
*University of Toronto*
*Vector Institute*

**Reviewed on OpenReview:** *https://openreview.net/forum?id=0zqt85NUyK*

## Abstract

Causal inference, especially in observational studies, relies on untestable assumptions about the true data-generating process. Sensitivity analysis helps us determine how robust our conclusions are when we alter these underlying assumptions. Existing frameworks for sensitivity analysis are concerned with worst-case changes in assumptions. In this work, we argue that using such pessimistic criteria can often become uninformative or lead to conclusions contradicting our prior knowledge about the world. To demonstrate this claim, we generalize the recent *s*-value framework (Gupta & Rothenhäusler, 2023) to estimate the sensitivity of three different common assumptions in causal inference. Empirically, we find that, indeed, worst-case conclusions about sensitivity can rely on unrealistic changes in the data-generating process. To overcome this, we extend the *s*-value framework with a new sensitivity analysis criterion: Bayesian Sensitivity Value (BSV), which computes the expected sensitivity of an estimate to assumption violations under priors constructed from real-world evidence. We use Monte Carlo approximations to estimate this quantity and illustrate its applicability in an observational study on the effect of diabetes treatments on weight loss.

## 1 Introduction

Causal inference offers powerful tools for decision-making across high-stakes domains, but both its internal and external validity rely on assumptions that cannot be verified from data alone (Bareinboim et al., 2022; Pearl & Bareinboim, 2011). Sensitivity analysis (Rosenbaum & Rubin, 1983a) has emerged as a critical tool to assess the robustness of causal conclusions to violations of these assumptions. Existing frameworks typically focus on the unconfoundedness assumption (VanderWeele & Ding, 2017; Zhao et al., 2019; Cinelli & Hazlett, 2020; Lu & Ding, 2023) — *all confounding variables have been measured and controlled for.* However, the internal validity of causal inference depends not only on unconfoundedness, but also on other model specification assumptions (Bang & Robins, 2005). As for external validity, even randomized trials are exposed to selection and outcome biases in their data collection mechanisms. For instance, participants included in a study may differ systematically from the broader population of interest, or certain outcomes may be reported only by a non-random subset of the study population.

The key question asked by a sensitivity analysis is: *What extent of violation of assumptions can an experiment tolerate, before its conclusions are reversed?* This is often useful when deciding whether a study provides strong enough evidence to make a decision, such as approving a drug for use. Further, practitioners are often interested in comparing subpopulations and prioritizing high-sensitivity ones when considering resource allocation and data collection for future experimentation (Razavi et al., 2021). However, since considering every possible combination of assumption violations is usually infeasible, common approaches estimate sensitivity in a worst-case sense and search for the smallest violation that reverses the study's conclusion (Cinelli & Hazlett, 2020; Gupta & Rothenhäusler, 2023). When applying these approaches to multi-dimensional settings, practitioners often consider smaller spaces of possible assumptions, *e.g.* one confounder at a time, a simplification that can render the analysis uninformative or even deceiving (Saltelli et al., 2019). Moreover, these practices can be entirely incompatible with real-world scenarios. First, worst-case violations may not be plausible in the real world, resulting in overly pessimistic results. Second, violations are likely to occur along multiple dimensions simultaneously, producing combinatorial effects that single-dimensional analyses fail to capture. Finally, trivially extending worst-case analyses to multi-dimensional settings can be particularly problematic as they tend to become uninformative as dimensionality increases.

In this work, we start by formalizing and showcasing the above challenges. To that end, we extend the *s*-value method of Gupta & Rothenhäusler (2023) to a more general sensitivity framework for causal inference. The *s*-value measures the minimum amount of shift in the covariate distribution that changes the sign of an average treatment effect estimate and hence reverses the corresponding causal conclusion. We generalize this to include other common assumptions in causal inference, including no unmeasured confounding, well-specified conditional models of outcomes given treatment and covariates, and external validity with respect to the distribution over covariates. This extended framework defines an assumption variable,

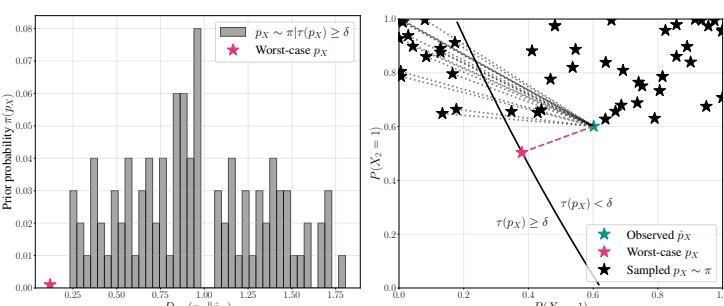

Figure 1: Let $\pi$ be a prior distribution over the covariate distribution $p_X$. The closest (worst-case) distribution over covariates that reverses the causal decision defined by a threshold $\delta$ has low probability under $\pi$, and more realistic distributions are further away from the observed and assumed $\hat{p}_X$ (Left). Further, samples from this prior that reverse the decision highlight sensitive distributions that are different from the worst-case one (Right).

the space of its possible values, a divergence metric to measure violations in this space, and constraints that enable practitioners to target particular sensitivity questions of interest.

Through simulation studies, we find that standard worst-case sensitivity analyses often provide limited actionable information when trying to distinguish between and prioritize directions in the space of an assumption's possible values. These approaches tend to be overly pessimistic because they treat all possible violations as equally likely and rarely reflect realistic violations. For instance, given our world knowledge, it is very unlikely to have a real-world shift both increasing family income and decreasing education level. Can we avoid attributing high sensitivity to an experiment based on such violations that contradict well-established knowledge of real populations? To address these limitations, we propose a Bayesian alternative, called the Bayesian Sensitivity Value (BSV), including its Empirical Bayes version (EBSV), which computes the expected sensitivity to assumption violations under a prior constructed from real-world evidence (data). Importantly, unlike existing Bayesian Causal Inference frameworks (McCandless et al., 2007; Li et al., 2023) that assume priors to compute posteriors for the treatment effect, BSV defines a posterior over changes in the data-generating process, while using existing established frequentist estimators. Further, EBSV incorporates data-driven priors instead of arbitrary ones. We review related frameworks in detail in section 5.

As an illustrative example, fig. 1 shows a toy setting to assess the sensitivity of a causal effect estimate with respect to the covariate distribution $p_X$, defined as the joint distribution over independent binary confounders $X_1$ and $X_2$. A worst-case analysis yields the closest distribution that reverses a causal decision based on a threshold $\delta$, despite having low probability under an empirical prior distribution. In contrast, distributions

sampled from an empirical prior that reverse this decision indicate lower sensitivity in magnitude and lie in a different direction in the space of possible assumptions on $p_X$. While the worst-case analysis might lead practitioners to direct resources towards distributions with small $P(X_1 = 1)$ and small $P(X_2 = 1)$, incorporating the empirical prior would instead prioritize those with small $P(X_1 = 1)$ and large $P(X_2 = 1)$.

Our work provides strategies to leverage large demographic and medical databases as well as detailed subgroup analyses of previous studies to construct empirical priors in a data-driven manner for a real-world diabetes study. The empirical priors allow us to analyze the sensitivity of an estimator to *realistic* violations of its assumptions. We provide practical algorithms to implement both, worst-case and Bayesian, sensitivity analyses. While worst-case analyses typically reduce to constrained optimization problems, Bayesian sensitivity computations reduce to a constrained sampling problem. Our empirical results demonstrate that sensitivity values in expectation differ substantially from worst-case scenarios, corroborating that conventional sensitivity analysis methods often fail to reflect realistic conditions. Moreover, we show how BSV can distinguish between directions in assumption space better than worst-case analyses, hence providing more informative guidance for critical experiment design decisions, such as defining appropriate study populations.

**Contributions.** The main contributions of our work are:

- We present a sensitivity analysis framework to unify different types of assumptions required by a given estimator, that allows practitioners to ask targeted questions about the sensitivity of a study.
- Our simulation studies and real-world application empirically demonstrate that worst-case analyses indeed often fail to reflect plausible scenarios, leading to overly pessimistic and uninformative results.
- We propose a new sensitivity criterion, the Bayesian Sensitivity Value (BSV), which is grounded in real-world evidence, is more realistic, and remains useful in high-dimensional spaces of assumptions.
- Finally, we illustrate BSV's ability to address the above limitations in our experiments, along with strategies to construct empirical priors from real-world evidence and practical algorithms for both criteria.

## 2 Sensitivity Analysis

To formalize our study of the effect of assumption violations on a causal decision, we will first define required notation and state three typical assumptions under which the popular outcome imputation estimator can identify average treatment effects. We then present our framework for sensitivity analysis with respect to each individual assumption in our notation, of which existing worst-case analyses are special cases.

### 2.1 Assumptions for Average Treatment Effect Identification

We consider the archetypal causal inference task: *binary treatment effect estimation*. In this task, we want to estimate the causal effect of a treatment relative to either another treatment or no treatment (control) in a population of interest. Then, we want to make a binary decision with respect to this estimate, *e.g.*, a treatment has a relative positive or negative effect. Concretely, consider a binary treatment variable $T \in \{0, 1\}$ and corresponding binary potential outcomes $Y(1), Y(0) \in \{0, 1\}$ under each treatment, as defined in the Neyman–Rubin model (Holland, 1986). In decision–focused causal inference Jaeschke et al. (1989); Hedayat et al. (2015), we translate estimates of the statistic called *average treatment effect* (ATE),

$$\tau := \mathbb{E}\big[Y(1) - Y(0)\big], \tag{1}$$

into binary choices such as $\mathbb{I}\big(\tau > \delta\big)$ for a user–specified threshold $\delta$. See a full list of notation in Appendix A.

The fundamental problem in causal inference is that we never observe the potential outcomes $(Y(0), Y(1))$. Instead, we observe the treatment $T$ and an outcome $Y = T \cdot Y(1) + (1 - T) \cdot Y(0)$. Therefore, different studies require assumptions of different forms and strengths to estimate the ATE consistently from observed data. Specifically, observational methods like inverse propensity score weighting (Horvitz & Thompson, 1952; Rosenbaum, 1987) and outcome regression (Rubin, 1977) convert eq. (1) into identifiable forms via a series of assumptions on its non-identifiable components. In this work, we treat these components as a collection of functions $a$, and parameterize the ATE as a function of these unknown functions: $\tau(a)$.

To illustrate this framework, consider the outcome imputation estimator. We start with the form of the ATE proposed in Lu & Ding (2023, Theorem 1) and then state the typical assumptions that make it identifiable from data. Given discrete covariates $X \in \mathcal{X}$, drawn from a distribution $p_X$ and observed jointly with treatment $T$ and outcome $Y$, we have

$$\tau(\varepsilon, \mu, p_X) = \sum_{x \in \mathcal{X}} p_X(x) \left[ \mathbb{E}\left[ \mu_1(x) \cdot \left( T + \frac{1-T}{\varepsilon_1(x)} \right) \,\middle|\, X = x \right] - \mathbb{E}\left[ \mu_0(x) \cdot \left( 1 - T + T\varepsilon_0(x) \right) \,\middle|\, X = x \right] \right]. \quad (2)$$

This equality holds, when the odds ratio function, $\varepsilon : \mathcal{X} \to \mathbb{R}_+^2$, is given by $\varepsilon(x) = (\varepsilon_0(x), \varepsilon_1(x))$ where

$$\varepsilon_0(x) = \frac{\mathbb{E}[Y(0)|T=1, X=x]}{\mathbb{E}[Y(0)|T=0, X=x]}, \quad \varepsilon_1(x) = \frac{\mathbb{E}[Y(1)|T=1, X=x]}{\mathbb{E}[Y(1)|T=0, X=x]},$$

the conditional outcome function, $\mu : \mathcal{X} \to [0,1]^2$, is given by $\mu(x) = (\mu_0(x), \mu_1(x))$ where

$$\mu_0(x) = \mathbb{E}[Y|T=0, X=x], \quad \mu_1(x) = \mathbb{E}[Y|T=1, X=x],$$

and the covariate distribution, $p_X \in \Delta^{|\mathcal{X}|-1}$ gives the probability $p_X(x)$. See Lu & Ding (2023) or appendix B for a proof of the equivalence between eqs. (1) and (2). The functions $(\varepsilon, \mu, p_X)$ are either non-identifiable from observational data or challenging to estimate. For this reason, causal inference makes assumptions on the possible values of $a = (\varepsilon, \mu, p_X)$. Below, we discuss each function and its typically assumed value.

The common **no unmeasured confounding** (Rosenbaum & Rubin, 1983b; Imbens & Rubin, 2015) assumption is that true odds ratios are identically 1, *i.e.*, $(\varepsilon_0(x), \varepsilon_1(x)) = (\mathbf{1}, \mathbf{1})$. Equivalently, potential outcomes are independent of treatment assignment, given observed covariates, *i.e.* $(Y(0), Y(1)) \perp\!\!\!\perp T|X$. Note that this assumption holds true in randomized experiments, but may be violated in observational studies due to unobserved confounders. Under the no unmeasured confounding assumption, eq. (2) reduces to

$$\tau(\mathbf{1}, \mu, p_X) = \sum_{x \in \mathcal{X}} p_X(x) \left[ \mu_1(x) - \mu_0(x) \right], \quad (3)$$

where the conditional outcomes are defined as before and $p_X$ is the target distribution of covariates $X$.

Assumptions of **well-specified conditional outcome models** (e.g., Rubin, 1977) typically state that the conditional outcome distributions $\mu$ can be learned from observed data by modeling the mapping from covariates and treatments to expected outcome, *i.e.*, the risk-minimizing models of outcomes $\mu^*$ are the true conditional outcome distributions $\mu$. This assumption may be violated due to model misspecification or reporting biases that modify the outcome distribution. Under no unmeasured confounding and correct specification of conditional outcome models, we now have

$$\tau(\mathbf{1}, \mu^*, p_X) = \sum_{x \in \mathcal{X}} p_X(x) \left[ \mu_1^*(x) - \mu_0^*(x) \right], \quad (4)$$

where $p_X$ remains the target distribution over covariates.

**External validity** (Shadish et al., 2002; Pearl & Bareinboim, 2011) assumptions typically state that the sampled population is the same as the target population of interest. In particular, in the context of a population distribution over covariates, an assumption of external validity would state that the sampled population's distribution over covariates $q_X$ is the same as the target distribution $p_X$. This assumption can be violated due to selection biases in data collection such that the observed population is not representative of the true target population. Under all three assumptions above, we have

$$\tau(\mathbf{1}, \mu^*, q_X) = \sum_{x \in \mathcal{X}} q_X(x) \left[ \mu_1^*(x) - \mu_0^*(x) \right] \approx \frac{1}{n} \sum_{i=1}^{n} \left[ \hat{\mu}_1(x^{(i)}) - \hat{\mu}_0(x^{(i)}) \right], \quad (5)$$

which is the standard and popular outcome imputation estimator, computed using a study population, $(x^{(i)}, t^{(i)}, y^{(i)})_{i=1}^{n}$, of $n$ individuals and (usually) fitted regression models $\hat{\mu}$ for conditional outcome distributions.

| Assumption | Function | Space $\mathcal{A}$ | Divergence $D$ | Typical assumption |
|---|---|---|---|---|
| Unconfoundedness | $\varepsilon$ | $\mathbb{R}_+^{2d}$ | Euclidean | $\mathbf{1}$ |
| Conditional Outcome Distributions | $\mu$ | $[0,1]^{2d}$ | KL | $\mu^*$ |
| Covariate Distribution | $p_X$ | $\Delta_{d-1}$ | KL | $q_X$ |

Table 1: Typical assumptions required for ATE identification.

While the example above corresponds to outcome imputation, it is straightforward to derive analogous parameterizations of assumptions for other estimators. As another example, we include the inverse propensity score weighting estimator in appendix C. Throughout this work, we present sensitivity criteria and algorithms with a focus on the setting with binary treatments and outcomes and discrete covariates. These may be applied to continuous-valued variables via discretization or continuous extensions that are left to future work.

Note that the three assumptions above are fundamentally distinct in that $p_X$ and $\mu$ are observable, though it may be challenging to access or represent all populations of interest, while $\varepsilon$ is inherently unobservable.

### 2.2 Our Sensitivity Analysis Framework

To quantify how violations of different assumptions affect causal decisions, we generalize the $s$-value of Gupta & Rothenhäusler (2023) with the following framework. For a given choice of assumption functions, $a = (\varepsilon, \mu, p_X)$, our goal is to quantify the impact of deviations from $a$ on the downstream decision function $\mathbb{I}\big(\tau(a) > \delta\big)$. For ease of interpretation, we will consider each of these functions one at a time, each of which can individually belong to very high-dimensional spaces. For any given function, we define:

  (i) $\mathcal{A}$ as a subset of possible triplets $(\varepsilon, \mu, p_X)$ over which possible values of that function range, and
  (ii) a divergence $D(a\|\hat{a})$ to quantify the degree of violation from an assumed value $\hat{a}$ if the correct one had been some $a \in \mathcal{A}$.

For binary outcomes and covariates that can take $d$ possible values ($|\mathcal{X}| = d$), the function of odds ratios $\varepsilon$ belong to $\mathbb{R}_+^{2d}$, conditionals $\mu$ are defined in $[0,1]^{2d}$, and the covariate distribution lies in the simplex $\Delta_{d-1}$. Hence, $\mathcal{A}$ respectively corresponds to the subsets $\{(\varepsilon, \mu^*, q_X)\}_{\varepsilon \in \mathbb{R}_+^{2d}}$, $\{(\mathbf{1}, \mu, q_X)\}_{\mu \in [0,1]^{2d}}$, and $\{(\mathbf{1}, \mu^*, p_X)\}_{p_X \in \Delta_{d-1}}$. We take $D$ as the Kullback-Leibler (KL) divergence for distributions $\mu$ and $p_X$ and Euclidean distance for the vectors $\varepsilon$. Table 1 summarizes these functions and their assumptions in our sensitivity framework.

Assume, without loss of generality, that under default values, $\tau\big(\mathbf{1}, \mu^*, q_X\big) > \delta$. Then, the region in $\mathcal{A}$ where $\tau(a) \leq \delta$ represents a reversal of the causal decision.

### 2.3 Worst-case Sensitivity Criterion

A special case of the framework described in section 2.2, the $s$-value, was introduced by Gupta & Rothenhäusler (2023) to measure the minimum amount of shift in the covariate distribution that changes the sign of an ATE estimate. In other words, they considered sensitivity to the $p_X$ function with decision threshold $\delta = 0$. Generalizing this $s$-value, we can characterize the worst-case sensitivity criterion for a study for any function.

**Definition 1** (Worst-case Sensitivity). *For a given function that has possible values in $\mathcal{A}$, its assumed value $\hat{a}$, and convex divergence measure $D$,* e.g. *Bregman divergences, if $\tau(\hat{a}) > \delta$, the worst-case sensitivity of a study to this assumption is defined as*

$$\sup_{a \in \mathcal{A}} \{\exp(-D(a\|\hat{a})) \mid \tau(a) \leq \delta\}. \tag{6}$$

Equation (6) ensures that the sensitivity values are bounded in $[0,1]$ for non-negative divergences and that higher values indicate higher sensitivity, since smaller deviations would be required to reverse the decision.

The analysis of odds ratio functions $\varepsilon$ by Lu & Ding (2023) can also be reinterpreted under this framework. Lu & Ding (2023) visualized the ATE estimates under different values of $\varepsilon_0$ and $\varepsilon_1$, assuming each to be constant in $X$, *i.e.* $\varepsilon \in \mathbb{R}_+^2$. Hence, $\mathcal{A}$ was restricted to be 2-dimensional. In their work, the visual gap

between $(1,1)$ and the *closest* $(\varepsilon_0, \varepsilon_1)$ that changed the sign of the ATE estimate was used to characterize sensitivity of the estimate to unobserved confounding. Our framework quantifies this worst-case sensitivity to the unconfoundedness assumption with a single scalar given by eq. (6). Importantly, it does not assume that $\varepsilon$ is a constant function of the covariates $X$, but instead belongs to $\mathbb{R}_+^{2d}$. Closely related is also the sensitivity analysis of Franks et al. (2020), with respect to the unconfoundedness assumption. This work defines sensitivity parameters $(\gamma_0, \gamma_1)$, which can be interpreted as the differences of expected conditional potential outcomes under the two treatments instead of the ratios $(\varepsilon_0, \varepsilon_1)$ used by Lu & Ding (2023), but are similarly treated as being constant in covariates $X$ for ease of visualization.

The absolute value of the scalar from eq. (6) is difficult to interpret. It is more practically useful to compare subsets of the $(\varepsilon, \mu, p_X)$ triplets that differ meaningfully, *e.g.* different $\mathcal{A}$ range over changes in selected subsets of the covariates. Then, each subset $\mathcal{A}$ is a collection of possible subpopulations varying in these covariates and practitioners may accordingly direct resources towards the $\mathcal{A}$ with high sensitivity. We describe constrained optimization algorithms to compute worst-case sensitivity values for all assumptions in section 4.1.

## 3   Bayesian Sensitivity Value

While sensitivity analyses are critical for decision-making under assumptions, they have not enjoyed widespread practical adoption (Tarantola et al., 2024) and have not always proved to be practically useful (Saltelli et al., 2019). In this work, we take a critical view of the worst-case sensitivity paradigm in particular, and highlight pitfalls that can render it uninformative or unrealistic in practical settings. The following real-world diabetes example is one such instance and motivates the need for alternate sensitivity criteria, beyond the worst-case.

**Motivating Example.**   Worst-case analyses compute the solution to the optimization problem in eq. (6) and we can compare this solution to empirical realizations of $a$ found in real populations. We did so in a diabetes application setting for the assumption on the joint distribution over binarized covariates, `Weight` and `HbA1c`, described in section 6.1. We also calculated the empirical distribution over the same variables in diabetes patients according to data collected by the National Health and Nutritional Examination Survey (National Center for Health Statistics, 2023, NHANES). Both distributions are visualized in fig. 2.

Note that the empirical distribution from the NHANES data puts highest probability on low `HbA1c` and low `Weight` values, in agreement with the well-established correlation between glycated hemoglobin reduction and weight loss (Gummesson et al., 2017). However, the worst-case analysis suggests sensitivity due to a distribution that puts large probability on low `HbA1c` and high `Weight` and is unlikely to appear in real diabetic populations. This exemplifies how worst-case analyses, once interpreted, may be based on optima that contradict known relationships between variables of interest and fail to reflect realistic scenarios.

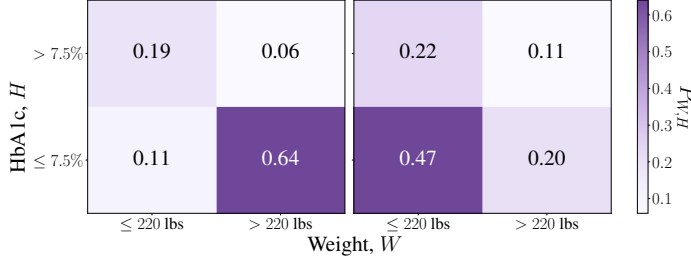

Figure 2: Optimum found by a worst-case analysis (Left) is significantly different from the real-world empirical distribution over the same variables recorded in the NHANES dataset (Right) and contradicts prior knowledge.

To tackle limitations of worst-case analyses, demonstrated above as well as in further experiments in section 6, we argue in favor of accounting for how likely any given assumption violation is in settings of interest. To this end, we treat the each of the functions $(\varepsilon, \mu, p_X)$ as random variables and propose a novel sensitivity criterion, called the Bayesian Sensitivity Value (BSV), defined as follows.

**Definition 2** (Bayesian Sensitivity Value). *Given an assumption $\hat{a} \in \mathcal{A}$, divergence $D$, and a random assumption $A$ taking values in $\mathcal{A}$ with distribution $\pi_A$, if $\tau(\hat{a}) > \delta$, the Bayesian Sensitivity Value is*

$$\mathbb{E}_{A \sim \pi_A}[\exp\left(-D(A\|\hat{a})\right) \mid \tau(A) \leq \delta]. \tag{7}$$

The BSV given by eq. (7) replaces the supremum of eq. (6) with an expectation computed with respect to a specified prior distribution over the assumption variable. This criterion has several advantages for sensitivity

analyses over its worst-case counterpart: (i) it is no more pessimistic than the worst-case sensitivity value, by definition, and far less pessimistic in practice,; (ii) the BSV is a direct result of the chosen prior $\pi_A$ and different values can be computed and interpreted under different choices of this prior; (iii) when available, this criterion allows practitioners to leverage large demographic, medical, or other domain-specific databases to construct evidence-based priors and compute an Empirical Bayesian Sensitivity Value (EBSV).

The EBSV grounds sensitivity analyses in real-world evidence and prior knowledge. For example, it is often possible to derive empirical distributions over common covariates from large demographic databases and surveys to construct a prior distribution over $p_X$. We can also use ATE estimates and subgroup analyses for different covariates and treatments from previous studies to construct priors over $\mu$. The resulting EBSV quantifies the sensitivity of a new study under the prior of previous findings. Empirically, we find that it also distinguishes between different subpopulations better than worst-case sensitivity values can, especially when assumptions can take values in high-dimensional spaces, as illustrated in section 6. We describe constrained sampling algorithms to compute Bayesian sensitivity values for all assumptions in section 4.2.

## 4  Practical Algorithms

Computation of the worst-case sensitivity value in eq. (6) and the BSV in eq. (7) requires solving constrained optimization and constrained sampling problems, respectively. Gupta & Rothenhäusler (2023) exploited the linearity of $\tau$ in $p_X$ to derive closed-form solutions for sensitivity to $p_X$. We can generalize their solutions by noticing that the constrained problem in eq. (6) is convex in each of $(\varepsilon, \mu, p_X)$ (or a one-to-one transformation of it), even if it isn't linear. We accordingly adapted existing optimization strategies via Lagrange multipliers to compute worst-case sensitivity values. The optimization algorithm presented here can be extended and applied to any causal estimand that can be written as a convex function of its assumption $a$, divergence $D$ that is convex in $a$, and convex assumption space $\mathcal{A}$. We used existing constrained sampling schemes for the BSV, though more efficient samplers may be derived by exploiting the convexity of eq. (7).

### 4.1  Constrained Optimization

We formulate eq. (6) as an equivalent minimization problem, with the following Langrangian.

$$\max_{\lambda} \min_{a \in \mathcal{A}} D(a\|\hat{a}) + \lambda(\tau(a) - \delta), \tag{8}$$

where $\lambda \geq 0$ is a Lagrange multiplier enforcing the threshold constraint on the ATE. We include any assumption-specific constraints on $a$, e.g. $\varepsilon_0(x) \geq 0, \varepsilon_1(x) \geq 0$ for all covariate sets $x$, with their own Lagrange multipliers. Due to its convexity, the above problem is solved exactly by an ascent-descent algorithm that alternates between updating the primal variable $a$ and performing gradient ascent on the dual variable $\lambda$ (Boyd & Vandenberghe, 2004; Boyd et al., 2011). For assumptions that represent probability distributions ($\mu$ and $p_X$), we used Entropic Mirror Descent (Beck & Teboulle, 2003) to solve the optimization problem under simplex constraints efficiently. For odds ratios $\varepsilon \in \mathbb{R}_+^{2d}$, we folded the non-negativity constraints into eq. (8) with its own Lagrange multipliers and used Gradient Descent to update $a$. We state the corresponding convex optimization problem for each assumption function and the standard dual ascent algorithm in appendix E.

### 4.2  Constrained Sampling

We compute a rejection-based Monte Carlo estimate (Robert et al., 1999) of the conditional expectation in eq. (7). Specifically, we drew samples from the given prior distribution, $A^{(i)} \sim \pi_A$ i.i.d., accepted them if they satisfy the constraint $\tau(A^{(i)}) \leq \delta$, and repeated this procedure until we obtain $M$ accepted samples, for some $M$. In all our experiments, we set $M = 5000$. Then, the BSV estimate is given by

$$\frac{1}{M} \sum_{i=1}^{N} \exp(-D(A^{(i)}|a)) \cdot \mathbb{I}\Big(\tau(A^{(i)}) \leq \delta\Big) \text{ where } N = \min\left\{ n \ \middle| \ \sum_{i=1}^{n} \mathbb{I}\Big(\tau(A^{(i)}) \leq \delta\Big) \geq M \right\}. \tag{9}$$

Hence, samples of the assumption parameters are accepted with the prior probability that they reverse the causal decision. Note that if this probability is very small, rejection sampling can be very inefficient. In this

case, more sophisticated sampling techniques like Markov Chain Monte Carlo methods (Gilks et al., 1995) may be adapted, though we found rejection sampling sufficient for our experiments.

Samples from the prior $\pi_A$ also allow us to estimate the probability of reversal of causal decisions due to assumption violations drawn from $\pi_A$, *i.e.* $\mathbf{P}_\pi(\tau(A) \leq \delta)$ via the empirical probability $\frac{1}{K} \sum_{i=1}^{K} \mathbb{I}\left(\tau(A^{(i)}) \leq \delta\right)$, for large $K = 1000000$. It is useful to jointly report the probability of decision reversal with the BSV and since the former accounts for how likely decision reversals are under the given prior and the latter provides information about the average "size" (via divergence $D$) of the assumption violation that reverses the decision. In our experiments, we additional report the acceptance rate $\frac{M}{N}$.

## 5 Related Work

Sensitivity analyses have long been recognized as critical for causal inference, particularly in observational studies (Rosenbaum, 1987). Traditional approaches, that typically consider worst-case violations of the unconfoundedness assumption, include Tan (2006); VanderWeele & Ding (2017); Cinelli & Hazlett (2020); Veitch & Zaveri (2020). Lu & Ding (2023) also proposed a flexible analysis for the unconfoundedness assumption that may be applied to different estimators. There are also Bayesian approaches to causal inference (Li et al., 2023) and sensitivity analysis (McCandless et al., 2007) in the context of this assumption. These works typically employ expert-elicited or uninformative priors. Gupta & Rothenhäusler (2023) tackled sensitivity to covariate distribution shifts and proposed the *s*-value framework, which we generalized to other assumptions, including those on conditional outcome distributions and unconfoundedness.

Causal sensitivity analyses have been adapted to several applications in fairness (Fawkes et al., 2024), optimization of operating characteristics of clinical trial design (Han et al., 2024) and evaluation of constrained selection biases (Cortes-Gomez et al., 2023). Recent critiques have highlighted limitations and misinterpretations associated with sensitivity analyses in practice, emphasizing a need for more realistic and informative methodologies (Ioannidis et al., 2019; Saltelli et al., 2019; de Souza et al., 2016). Our work takes a step towards addressing these limitations by grounding sensitivity analyses in real-world evidence.

Several large databases, collected and curated nationally or globally, provide empirical data on demographic and medical variables (Clore et al., 2014; CDC, 2023). Additionally, post-hoc analyses of subgroup effects are commonly found in medical literature (Tchang et al., 2025; Kadowaki et al., 2024; Ma et al., 2024; Ryan et al., 2024). Our proposed Bayesian Sensitivity Value provides a way to leverage these sources of information to understand sensitivity with respect to $p_X$ and $\mu_X$, respectively, while representing real-world populations.

## 6 Empirical Analysis

Our experiments use sensitivity analyses to compare and rank subsets of the $(\varepsilon, \mu, p_X)$ triplets, denoted $\mathcal{A}$, that differ meaningfully, so that high-sensitivity subgroups may be discovered, for instance, for resource prioritization. We computed worst-case sensitivity value, the BSV under a uniform or uninformative prior, and the EBSV under evidence-based priors, when they are available, to investigate the following questions: (i) How do the different sensitivity analysis criteria scale with dimensionality of the space of possible assumptions $\mathcal{A}$, specifically in their ability to distinguish between different $\mathcal{A}$? (ii) Do BSV analyses, including EBSV, differ meaningfully from worst-case analyses, specifically to uncover different sensitivity rankings across $\mathcal{A}$? (iii) How does BSV perform in real-world applications, compared to worst-case analyses?

### 6.1 Experimental Settings

**Simulation.** We constructed a simulation study to generate data with binary covariates and heterogeneous treatment effects, where the true treatment effect varies across subgroups defined by covariate values. Observational data was generated to reflect realistic biases commonly encountered in practice: confounding through treatment assignment and selection bias based on outcomes, treatment assignment, and covariates. We used settings with varying numbers of covariates $(4, 6, 8)$ to explore how the dimensionality of the problem affects sensitivity analyses. See full details of the data generation in appendix D.1. For BSV computations,

we considered an idealistic setting and constructed empirical prior distributions for $p_X$ and $\mu$ using samples from the true data-generation mechanism without confounding or selection biases.

**Diabetes Application.** As a real-world application, we adapted the Semaglutide vs. Tirzepatide dataset and large-language-model (LLM) based causal estimator from Dhawan et al. (2024). This study leveraged patient-reported data from relevant subreddits, employed large language models to extract observational distributions, accounted for confounding due to `Age`, `Sex`, `BMI` (Body Mass Index), `HbA1c` (Glycated Hemoglobin), and `Weight`, and estimated the relative effect of diabetes treatments on a weight loss outcome. Given the potential selection biases in both the data collection and modeling via LLMs, informative sensitivity analyses are crucial for reliable inference in this setting. See appendix D.2 for complete details.

This setting is also useful to demonstrate the construction of empirical priors and practical instantiation of an EBSV analysis when true priors are unavailable. Here, we leveraged previous databases and studies for information relevant to the assumptions of interest, $p_X$ and $\mu$. Specifically, we extracted distributions over `Age`, `Sex`, `HbA1c`, and `Weight` across different races using the Diabetes dataset from the UCI repository (Clore et al., 2014) and used them to fit a Dirichlet distribution that serves as a prior over the covariate distribution $p_X$. We obtained distributions over the conditional outcome distributions for Semaglutide ($T = 0$) from subgroup analyses conducted in previous studies (Kadowaki et al., 2024) and similarly fit a Dirichlet distribution to use as the empirical prior. We note that constructing empirical priors on odds ratios $\varepsilon$ from data remains a challenge since they are inherently unobserved quantities. Hence, we use user-defined priors which are, in this case, truncated Gaussians, centered at **1**, with scales $\sigma \in \{1.0, 1.5\}$.

**Comparing different $\mathcal{A}$.** Following the comparison of sensitivity values across covariates in Gupta & Rothenhäusler (2023), we constructed comparisons between different $\mathcal{A}$ for all our assumptions, where possible elements of each $\mathcal{A}$ range over particular covariates and treatments. Concretely, this corresponds to calculating sensitivity values when elements of $\mathcal{A}$ differ in their values of either the odds ratios $\varepsilon_{X_j}$ for covariates $X_j$, outcome distributions $\mu_{t,X_j} = \mathbb{E}[Y|T = t, X_j]$ conditioned on a treatment $t$ and covariates $X_j$, or distributions $p_{X_j}$ over covariates $X_j$. Here, $X_j$ is any subset of all the covariates $X$, and including more covariates results in higher dimensional $\mathcal{A}$. For example, to find the pair of binary covariates whose joint distribution $\tau(\varepsilon, \mu, p_X)$ is most sensitive to, we compared sensitivity values given by eq. (6) or eq. (7) with

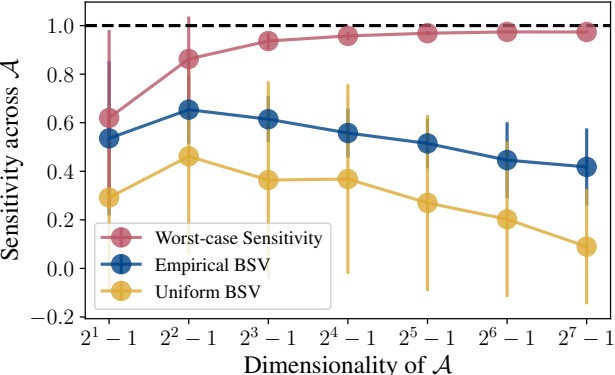

Figure 3: For high-dimensional $\mathcal{A}$, worst-case sensitivity values for covariate distributions are all close to 1 (mean $\approx 1$, variance $\approx 0$), while BSVs, both Uniform and Empirical, distinguish between different $\mathcal{A}$ better.

$\mathcal{A} = \{(\mathbf{1}, \mu^*, q_{X_{-j}}, p_{X_j})\}_{p_{X_j} \in \Delta_3}$ for different pairs of covariates $X_j$, where $X_{-j}$ includes all other covariates. This results in a sensitivity ranking according to either the worst-case or the Bayesian criterion.

## 6.2 Simulations

To study the behavior of sensitivity criteria at scale and answer item (i), we simulated a setting with eight binary covariates and measured sensitivity to the covariate distribution where $\mathcal{A}$ has increasing dimensionality. For $k \in \{1, \ldots, 7\}$, we considered every possible subset of covariates of size $k$, set $\mathcal{A}$ to be the space of possible joint distributions over this covariate subset, which is $\Delta_{2^k-1}$, and solved for the worst-case and Bayesian sensitivity values in each case. Figure 3 shows the mean and standard deviation of these sensitivity values against the dimensionality of $\mathcal{A}$. As dimensionality increases, worst-case sensitivity values quickly tend toward a mean of 1 and standard deviation of 0, hence attributing highest possible sensitivity to all high-dimensional $\mathcal{A}$, regardless of the plausibility of the worst-case violation in practice. In contrast, the BSV does not appear to suffer from this curse of dimensionality in our settings, maintaining variability across different $\mathcal{A}$ even in high dimensions and yielding a sensitivity ranking that may be used to inform resource allocation in practice.

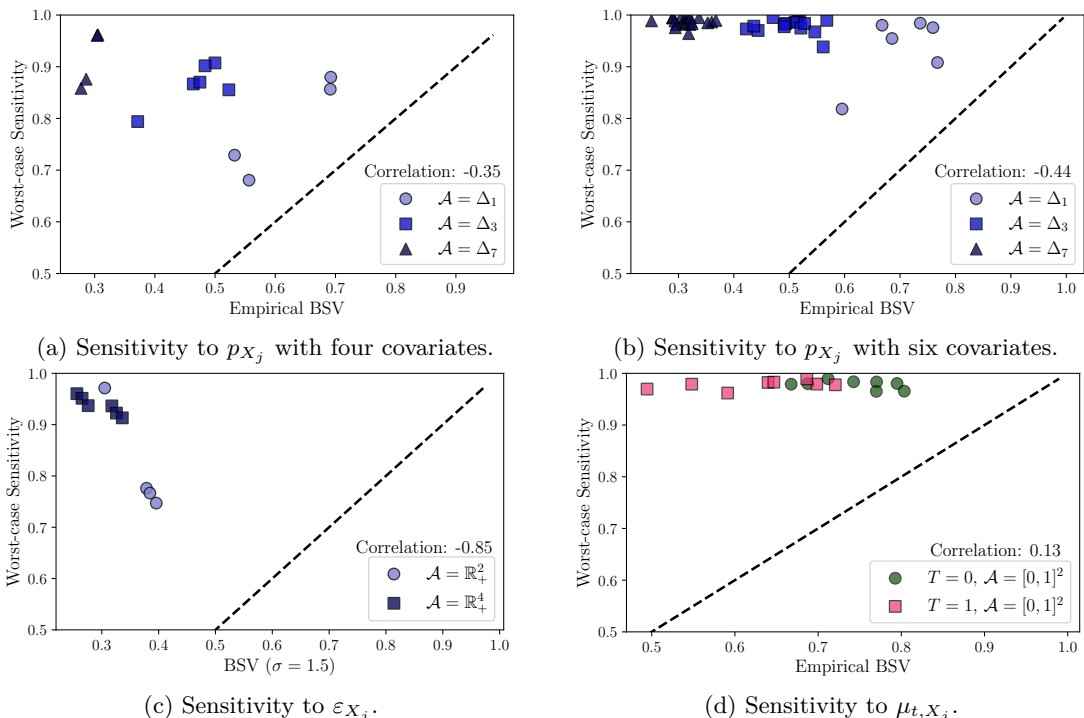

Figure 4: For all assumptions and different choices of subsets $\mathcal{A}$, Bayesian sensitivity is uncorrelated or negatively correlated with worst-case sensitivity, revealing different trends in sensitivity rankings.

Next, we investigated the actual sensitivity rankings produced by BSV versus a worst-case analysis to answer item (ii). Figure 4 shows the correlation between these rankings for assumptions on $\varepsilon_{X_j}$, $\mu_{t,X_j}$, and $p_{X_j}$ (with four or six total covariates) for different $X_j$ and $t$. Again, worst-case analyses are relatively pessimistic, assigning very similar and high sensitivity values, especially when $\mathcal{A}$ is high-dimensional. Moreover, the worst-case sensitivity rankings are uncorrelated or even negatively correlated with BSV rankings. Hence, BSV produces different trends in sensitivity across $\mathcal{A}$. Sensitivity values for different $X_j$ and $t$ under the worst-case, empirical BSV, and BSV with uniform priors are presented in figs. 6 to 8 of appendix F.

For some intuition on this behavior, consider the space of assumption violations. Adding more dimensions to this space allows violations in more variables. This is likely to increase the worst-case sensitivity, as the feasible region of the constrained optimization problem becomes larger. In contrast, adding dimensions to the space of assumption violations in the constrained sampling problem that yields the BSV also adds dimensions to the prior distribution $\pi$. Empirical joint priors over more variables can effectively constrain the space of *likely* assumption violations, especially when these variables are correlated with each other. Sampling from this prior results in BSVs that account for the choice of variables and provides more meaningful trends.

## 6.3 Diabetes application

We conducted similar sensitivity analyses under the worst-case criterion, the BSV with uniform priors, and the empirical BSV, on the diabetes study, with their results visualized in fig. 5. Here, we consider individual covariates as well as `Covariate pair indices`, which lexicographically index all possible pairs of covariates in {`Age`, `Sex`, `BMI`, `HbA1c`, `Weight`}. In the analyses with respect to $p_X$ and $\mu$, shown in figs. 5b and 5c respectively, worst-case values and BSV always agree in zero-sensitivity settings as expected. However, BSV highlights informative trends in others, which can support targeted future experimentation and data collection. Notably, they suggest that the estimated ATE is less sensitive than what a worst-case analysis would indicate in the `Weight > 220 lbs` subpopulation. Similar to the simulation study, fig. 5a shows BSV for odds ratios under truncated Gaussian priors. While differences are small between choices of observed covariates, sensitivity is lower whenever `Weight` is among them. Further, table 2 lists the sensitivity values given by worst-case, uniform Bayesian, and empirical Bayesian criteria, along with acceptance rates and probabilities of decision

Table 2: In a real-world diabetes experiment, the probabilities of decision reversal $\mathbf{P}_\pi(\tau(A) \leq \delta)$ and corresponding BSVs under uniform and empirical priors indicate how likely a decision reversal is and the average size of assumption violation required for reversal, respectively. Here, $\mathbf{AR}$ = acceptance rate.

| | Worst-case | Uniform BSV (AR) | $\mathbf{P}_\mathcal{U}(\tau(A) \leq \delta)$ | (E)BSV (AR) | $\mathbf{P}_\pi(\tau(A) \leq \delta)$ |
|---|---|---|---|---|---|
| **Covariate distributions $p_X$** | | | | | |
| Age | 0.00 | 0.00 (0.00) | 0.00 | 0.00 (0.00) | 0.00 |
| Sex | 0.00 | 0.00 (0.00) | 0.00 | 0.00 (0.00) | 0.00 |
| HbA1c | 0.00 | 0.00 (0.00) | 0.00 | 0.00 (0.00) | 0.00 |
| Weight | 0.52 | 0.30 (0.57) | 0.58 | 0.14 (1.00) | 0.99 |
| Age, Sex | 0.00 | 0.00 (0.00) | 0.00 | 0.00 (0.00) | 0.00 |
| Age, HbA1c | 0.00 | 0.00 (0.00) | 0.00 | 0.00 (0.00) | 0.00 |
| Age, Weight | 0.57 | 0.31 (0.57) | 0.57 | 0.10 (1.00) | 1.00 |
| Sex, HbA1c | 0.00 | 0.00 (0.00) | 0.00 | 0.00 (0.00) | 0.00 |
| Sex, Weight | 0.64 | 0.27 (0.50) | 0.51 | 0.08 (1.00) | 1.00 |
| HbA1c, Weight | 0.66 | 0.29 (0.48) | 0.49 | 0.09 (1.00) | 1.00 |
| **Conditional outcome distributions $\mu$** | | | | | |
| Age$\leq$ 45, $T = 0$ | 0.97 | 0.97 (0.06) | 0.06 | 0.94 (0.38) | 0.37 |
| Age$>$ 45, $T = 0$ | 0.97 | 0.97 (0.03) | 0.03 | 0.95 (0.66) | 0.67 |
| Age$\leq$ 45, $T = 1$ | 0.88 | 0.48 (0.90) | 0.89 | 0.42 (0.83) | 0.83 |
| Age$>$ 45, $T = 1$ | 0.95 | 0.33 (0.95) | 0.95 | 0.30 (0.88) | 0.88 |
| Sex=M, $T = 0$ | 0.96 | 0.97 (0.06) | 0.06 | 0.95 (0.37) | 0.37 |
| Sex=F, $T = 0$ | 0.96 | 0.97 (0.03) | 0.03 | 0.95 (0.41) | 0.41 |
| Sex=M, $T = 1$ | 0.84 | 0.46 (0.91) | 0.91 | 0.40 (0.84) | 0.84 |
| Sex=F, $T = 1$ | 0.85 | 0.32 (0.95) | 0.95 | 0.30 (0.88) | 0.88 |
| BMI$\leq$ 28.5, $T = 0$ | 0.98 | 0.97 (0.08) | 0.08 | 0.94 (0.51) | 0.50 |
| BMI$>$ 28.5, $T = 0$ | 0.99 | 0.98 (0.04) | 0.04 | 0.96 (0.29) | 0.29 |
| BMI$\leq$ 28.5, $T = 1$ | 0.95 | 0.48 (0.92) | 0.92 | 0.51 (0.75) | 0.76 |
| BMI$>$ 28.5, $T = 1$ | 0.97 | 0.40 (0.96) | 0.96 | 0.32 (0.90) | 0.89 |
| HbA1c$\leq$ 7.5, $T = 0$ | 0.96 | 0.97 (0.06) | 0.06 | 0.94 (0.52) | 0.52 |
| HbA1c$>$ 7.5, $T = 0$ | 0.96 | 0.97 (0.03) | 0.03 | 0.96 (0.12) | 0.12 |
| HbA1c$\leq$ 7.5, $T = 1$ | 0.85 | 0.46 (0.91) | 0.91 | 0.41 (0.84) | 0.84 |
| HbA1c$>$ 7.5, $T = 1$ | 0.84 | 0.32 (0.95) | 0.94 | 0.29 (0.88) | 0.88 |
| Weight$\leq$ 220, $T = 0$ | 0.00 | 0.00 (0.00) | 0.00 | 0.00 (0.00) | 0.00 |
| Weight$>$ 220, $T = 0$ | 1.00 | 0.56 (1.00) | 1.00 | 0.75 (1.00) | 1.00 |
| Weight$\leq$ 220, $T = 1$ | 0.30 | 0.10 (0.77) | 0.76 | 0.08 (0.70) | 0.71 |
| Weight$>$ 220, $T = 1$ | 1.00 | 0.76 (0.52) | 0.51 | 0.82 (0.51) | 0.51 |
| **Odds ratio functions $\varepsilon$** | | | | | |
| Age | 0.98 | — | — | 0.61 (0.37) | 0.37 |
| Sex | 0.98 | — | — | 0.61 (0.38) | 0.38 |
| BMI | 0.99 | — | — | 0.61 (0.38) | 0.39 |
| HbA1c | 0.98 | — | — | 0.61 (0.38) | 0.38 |
| Weight | 0.98 | — | — | 0.59 (0.42) | 0.42 |
| Age, Sex | 0.99 | — | — | 0.57 (0.31) | 0.32 |
| Age, BMI | 0.99 | — | — | 0.58 (0.30) | 0.31 |
| Age, HbA1c | 0.99 | — | — | 0.57 (0.31) | 0.32 |
| Age, Weight | 0.99 | — | — | 0.56 (0.35) | 0.35 |
| Sex, BMI | 0.99 | — | — | 0.57 (0.31) | 0.31 |
| Sex, HbA1c | 0.99 | — | — | 0.56 (0.36) | 0.36 |
| Sex, Weight | 0.99 | — | — | 0.56 (0.36) | 0.36 |
| BMI, HbA1c | 0.99 | — | — | 0.57 (0.32) | 0.32 |
| BMI, Weight | 0.99 | — | — | 0.56 (0.36) | 0.36 |
| HbA1c, Weight | 0.99 | — | — | 0.56 (0.36) | 0.36 |

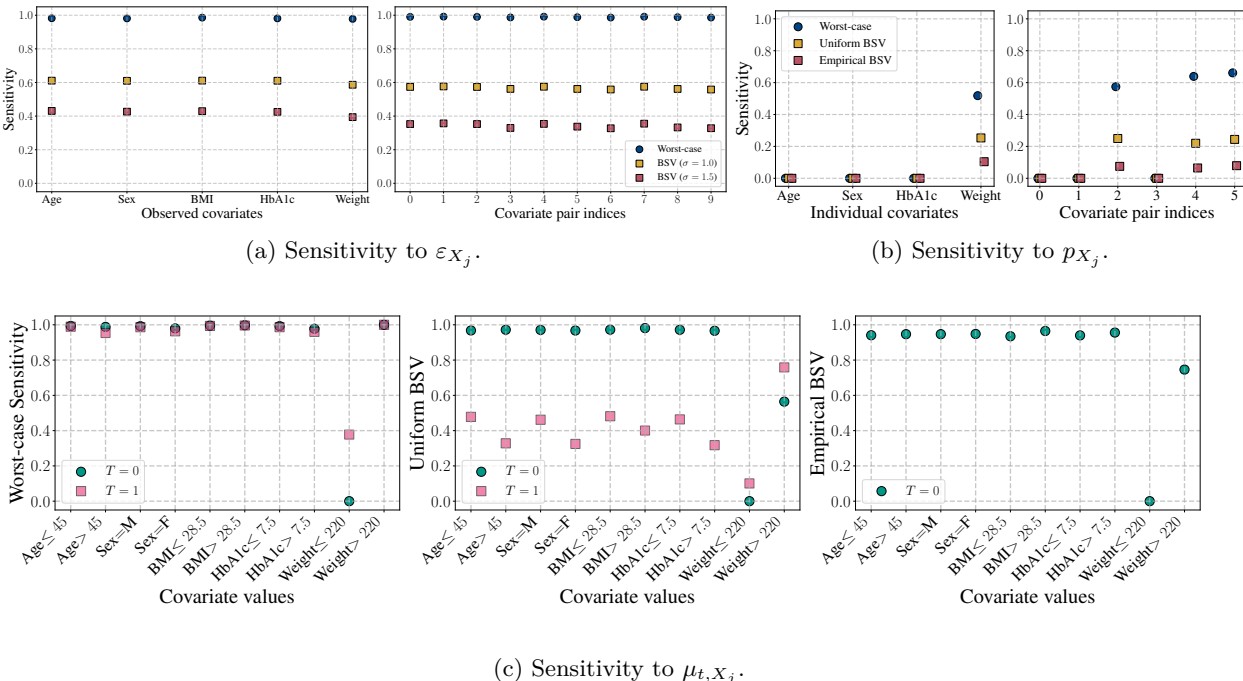

(a) Sensitivity to $\varepsilon_{X_j}$.

(b) Sensitivity to $p_{X_j}$.

(c) Sensitivity to $\mu_{t,X_j}$.

Figure 5: BSV for odds ratios in fig. 5a, covariate distributions in fig. 5b, and conditional outcome distributions fig. 5c in a real-world diabetes experiment can be more informative than worst-case analyses. In particular, BSV suggests relatively low sensitivity in the `Weight > 220 lbs` subpopulation, as compared to the worst-case analysis which assigns it the highest possible sensitivity value.

reversal under uniform and empirical priors. Notably, a small BSV and large probability of decision reversal indicate that decision reversals are likely under the given prior but the corresponding assumption violations are far from the observed or assumed value of $A$, as measured by divergence $D$.

## 7 Limitations

While this work has taken a critical view of worst-case sensitivity analyses, the proposed Bayesian Sensitivity Value is not without limitations in its real-world application. (i) Our experiments suggest that the empirical BSV under data-driven priors can be practically useful. However, constructing such priors on odds ratios $\varepsilon$ from data remains fundamentally challenging since they are inherently unobservable quantities. (ii) We considered and compared different subsets $\mathcal{A}$ of possible assumptions that were of the same form and admitted the same divergence metric, *e.g.* KL divergence for distributions lying in a simplex. However, combining different spaces of possible assumptions, along with their corresponding divergence metrics may allow practitioners to ask more interesting questions about sensitivity and capture interactions between different assumptions. (iii) The practical implementation of BSV computations is a simple rejection-based sampling technique, which can be very inefficient for small prior probability of reversing a causal decision. More involved and efficient techniques would help improve the adoption of this sensitivity criterion in practice. (iv) Finally, while the BSV captures the variation and plausibility of assumptions under a given prior, this also means that it is highly prior-dependent and may be susceptible to prior misspecification.

## 8 Conclusions

In this work, we generalized the *s*-value framework to three common assumptions in causal inference and proposed a new sensitivity criterion, the Bayesian Sensitivity Value, that can leverage real-world evidence. Our empirical studies demonstrated that worst-case sensitivity analyses can be overly pessimistic and uninformative, especially for high-dimensional spaces of possible assumptions, and rarely reflect realistic

shifts in assumptions. However, the BSV distinguished between different $\mathcal{A}$ even in high-dimensional spaces and revealed trends that differ from worst-case analyses. There are several exciting directions for future research: (i) constructing informative priors for different assumptions, including odds ratios, using varied sources of real-world evidence, (ii) adapting more sophisticated sampling techniques than rejection sampling to improve efficiency of BSV computations, and (iii) deriving mathematically rigorous strategies to operationalize sensitivity analyses for future experiment design. Given the critical importance of sensitivity analyses for causal decision-making in high-stakes domains like medicine, it is essential to improve their practically utility. While traditional analyses have typically focused on worst-case violations of assumptions, the BSV is a step towards a more informative criterion for real-world applications and improved resource prioritization.

**Broader Impact Statement**

Any sensitivity analysis framework has important societal implications for causal inference in high-stakes domains like healthcare. While the BSV can improve decision-making by providing more realistic sensitivity analyses and a better understanding of treatment effects across subpopulations, it also carries risks: the quality of analyses depends critically on the representativeness of real-world evidence used to construct priors, and the method could be misused to justify decisions that appear more robust than they actually are. To mitigate these risks, we recommend transparent documentation of prior construction and use of multiple sensitivity criteria, including our Bayesian criterion as well as traditional worst-case analyses.

**Acknowledgments**

We would like to thank Ayoub El Hanchi for helpful discussions and feedback on the paper. Resources used in preparing this research were provided in part by the Province of Ontario, the Government of Canada through CIFAR, and companies sponsoring the Vector Institute. We acknowledge the support of the Natural Sciences and Engineering Research Council of Canada (NSERC), RGPIN-2021-03445.

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

# A Notation

| | |
|---|---|
| $A$ | Assumption random variable whose value must be assumed for causal inference. |
| $\pi_A$ | Prior distribution over assumption variable $A$. |
| $X$ | Random variable corresponding to features of an individual in a causal inference dataset. |
| $T$ | Random variable corresponding to treatment or intervention assigned to an individual in a causal inference dataset. |
| $Y$ | Random variable corresponding to outcome observed for an individual in a causal inference dataset. |
| $x$ | Possible instance of $X$ from its support $\mathcal{X}$. |
| $t$ | Possible instance of $T$ from its support $\mathcal{T} = \{0, 1\}$ (binary treatments). |
| $y$ | Possible instance of $Y$ from its support $\mathcal{Y} = \{0, 1\}$ (binary outcomes). |
| $a$ | Possible instance or sampled value of $A$ from its support $\mathcal{A}$. |
| $Y(t)$ | Random variable corresponding to potential outcome observed for an individual after receiving treatment $t$. |
| $\varepsilon(X)$ | Unconfoundedness assumption parameters, $(\varepsilon_0(X), \varepsilon_1(X))$. |
| $\mu(X)$ | Conditional outcome distributions given covariates and treatment, $(\mu_0(X), \mu_1(X))$. |
| $p_X$ | Distribution over covariates $X$. |
| $x^{(i)}$ | Sampled value of $X$ for individual $i$. |
| $t^{(i)}$ | Sampled value of $T$ for individual $i$. |
| $y^{(i)}$ | Sampled value of $Y$ for individual $i$. |
| $\tau$ | Average treatment effect (ATE) given by $\mathbb{E}[Y(1) - Y(0)]$, where the expectation is over some defined population of individuals. |
| $\tau(a)$ | ATE computed or estimated under the assumption $a$. |
| $\delta$ | Decision threshold on the (estimated) ATE to choose between treatments. |
| $n$ | Total number of individuals. |

## B  ATE Identification Proof

We reproduce a proof of the result in Lu & Ding (2023) for eq. (2), in our notation, for completeness.

$$
\begin{aligned}
\tau &:= \mathbb{E}\big[Y(1) - Y(0)\big] \\
&= \mathbb{E}\big[Y(1) \mid T = 1\big]P(T = 1) + \mathbb{E}\big[Y(1) \mid T = 0\big]P(T = 0) \\
&\quad - \mathbb{E}\big[Y(0) \mid T = 1\big]P(T = 1) - \mathbb{E}\big[Y(0) \mid T = 0\big]P(T = 0), \\
&= \mathbb{E}\big[Y \mid T = 1\big]P(T = 1) + \mathbb{E}\big[Y(1) \mid T = 0\big]P(T = 0) \\
&\quad - \mathbb{E}\big[Y(0) \mid T = 1\big]P(T = 1) - \mathbb{E}\big[Y \mid T = 0\big]P(T = 0),
\end{aligned}
$$

where the last equality follows from the definition of $Y$.

For $\mu_0(X) = \mathbb{E}[Y|T = 0, X]$ and $\mu_1(X) = \mathbb{E}[Y|T = 1, X]$,

$$
\begin{aligned}
\mathbb{E}\big[Y \mid T = 1\big]P(T = 1) &= \mathbb{E}\big[\mathbb{E}\big[Y \mid T = 1, X\big] \mid T = 1\big]P(T = 1) \\
&= \mathbb{E}\big[\mu_1(X) \mid T = 1\big]P(T = 1). \\
\mathbb{E}\big[Y \mid T = 0\big]P(T = 0) &= \mathbb{E}\big[\mathbb{E}\big[Y \mid T = 0, X\big] \mid T = 0\big]P(T = 0) \\
&= \mathbb{E}\big[\mu_0(X) \mid T = 0\big]P(T = 0).
\end{aligned}
$$

For $\varepsilon_0(X) = \dfrac{\mathbb{E}[Y(0)|T = 1, X]}{\mathbb{E}[Y(0)|T = 0, X]}$ and $\varepsilon_1(X) = \dfrac{\mathbb{E}[Y(1)|T = 1, X]}{\mathbb{E}[Y(1)|T = 0, X]}$,

$$
\begin{aligned}
\mathbb{E}\big[Y(1) \mid T = 0\big]P(T = 0) &= \mathbb{E}\big[\mathbb{E}\big[Y(1) \mid T = 0, X\big] \mid T = 0\big]P(T = 0) \\
&= \mathbb{E}\left[\frac{\mu_1(X)}{\varepsilon_1(X)} \mid T = 0\right]P(T = 0). \\
\mathbb{E}\big[Y(0) \mid T = 1\big]P(T = 1) &= \mathbb{E}\big[\mathbb{E}\big[Y(0) \mid T = 1, X\big] \mid T = 1\big]P(T = 1) \\
&= \mathbb{E}\big[\mu_0(X)\varepsilon_0(X) \mid T = 1\big]P(T = 1).
\end{aligned}
$$

Plugging the above back into the equation for $\tau$, we have,

$$
\begin{aligned}
\tau &= \mathbb{E}\big[Y \mid T = 1\big]P(T = 1) + \mathbb{E}\big[Y(1) \mid T = 0\big]P(T = 0) \\
&\quad - \mathbb{E}\big[Y(0) \mid T = 1\big]P(T = 1) - \mathbb{E}\big[Y \mid T = 0\big]P(T = 0) \\
&= \mathbb{E}\big[\mu_1(X) \mid T = 1\big]P(T = 1) + \mathbb{E}\left[\frac{\mu_1(X)}{\varepsilon_1(X)} \mid T = 0\right]P(T = 0) \\
&\quad - \mathbb{E}\big[\mu_0(X)\varepsilon_0(X) \mid T = 1\big]P(T = 1) - \mathbb{E}\big[\mu_0(X) \mid T = 0\big]P(T = 0) \\
&= \mathbb{E}\left[\mu_1(X) \cdot \left(T + \frac{1 - T}{\varepsilon_1(X)}\right)\right] - \mathbb{E}\left[\mu_0(X) \cdot \big(T\varepsilon_0(X) + 1 - T\big)\right],
\end{aligned}
$$

which is equivalent to eq. (2) by the law of iterated expectations.

# C   Assumptions for Inverse Propensity Score Weighting

Consider the inverse propensity score weighting estimator. Equivalent to the form of the ATE proposed in Lu & Ding (2023, Theorem 2), we have

$$\tau(\varepsilon, e, p_{XTY}) = \sum_{\substack{x \in \mathcal{X} \\ (t,y) \in \{0,1\}^2}} p_{XTY}(x,t,y) \left[ w_1(x) \cdot \left( \frac{ty}{e(x)} \right) - w_0(x) \cdot \left( \frac{(1-t)y}{1-e(x)} \right) \right], \qquad (10)$$

where $w_1(x) = e(x) + \frac{1-e(x)}{\varepsilon_1(x)}$ and $w_0(x) = e(x)\varepsilon_0(x) + 1 - e(x)$.

This equality holds, when the odds ratio function, $\varepsilon : \mathcal{X} \to \mathbb{R}_+^2$, is given by $\varepsilon(x) = (\varepsilon_0(x), \varepsilon_1(x))$ where

$$\varepsilon_0(x) = \frac{\mathbb{E}[Y(0)|T=1, X=x]}{\mathbb{E}[Y(0)|T=0, X=x]}, \quad \varepsilon_1(x) = \frac{\mathbb{E}[Y(1)|T=1, X=x]}{\mathbb{E}[Y(1)|T=0, X=x]},$$

the propensity scores, $e : \mathcal{X} \to [0,1]^2$, are given by

$$e(x) = \mathbb{E}[T|X=x],$$

and the joint distribution, $p_{XTY} \in \Delta^{4|\mathcal{X}|-1}$ gives the probability $p_{XTY}(x,t,y)$.

The functions $(\varepsilon, \mu, p_{XTY})$ are either non-identifiable from observational data or challenging to estimate. So we must make the following assumptions on the possible values of $\mathbf{a} = (\varepsilon, \mu, p_{XTY})$.

Once again, the **no unmeasured confounding** assumption is that true odds ratios are identically 1, *i.e.*, $(\varepsilon_0(x), \varepsilon_1(x)) = (\mathbf{1}, \mathbf{1})$, as discussed in section 2. Under this assumption, eq. (10) reduces to

$$\tau(\mathbf{1}, e, p_{XTY}) = \sum_{\substack{x \in \mathcal{X} \\ (t,y) \in \{0,1\}^2}} p_{XTY}(x,t,y) \left[ \frac{ty}{e(x)} - \frac{(1-t)y}{1-e(x)} \right], \qquad (11)$$

where the propensity scores $e$ and jont distribution $p_{XTY}$ are defined as before.

Assumptions of **well-specified propensity score models** (e.g., Horvitz & Thompson, 1952) typically state that the propensity scores $e$ can be learned from observed data by modeling the mapping from covariates to treatments, *i.e.*, the risk-minimizing models of outcomes $e^*$ are the true propensity scores $e$. This assumption may be violated due to model misspecification or reporting biases that modify the treatment assignment distribution. Under no unmeasured confounding and correct specification of propensity score models, we have

$$\tau(\mathbf{1}, e^*, p_{XTY}) = \sum_{\substack{x \in \mathcal{X} \\ (t,y) \in \{0,1\}^2}} p_{XTY}(x,t,y) \left[ \frac{ty}{e^*(x)} - \frac{(1-t)y}{1-e^*(x)} \right]. \qquad (12)$$

**External validity of** $p_{XTY}$ assumptions typically state that the sampled population is the same as the target population of interest. Here, an assumption of external validity would state that the sampled population's distribution over covariates, treatments, outcomes $q_{XTY}$ is the same as the target distribution $p_{XTY}$. This assumption can be violated due to selection biases in data collection such that the observed population is not representative of the true target population. Under all three assumptions above, we have

$$\tau(\mathbf{1}, e^*, q_{XTY}) = \sum_{\substack{x \in \mathcal{X} \\ (t,y) \in \{0,1\}^2}} q_{XTY}(x,t,y) \left[ \frac{ty}{e^*(x)} - \frac{(1-t)y}{1-e^*(x)} \right] \qquad (13)$$

$$\approx \frac{1}{n} \sum_{i=1}^{n} \left[ \frac{t^{(i)}y^{(i)}}{\hat{e}(x^{(i)})} - \frac{(1-t^{(i)})y^{(i)}}{1-\hat{e}(x^{(i)})} \right], \qquad (14)$$

which is the standard and popular inverse propensity weighting estimator, computed using a study population, $(x^{(i)}, t^{(i)}, y^{(i)})_{i=1}^n$, of $n$ individuals and (usually) fitted regression models $\hat{e}$ for propensity scores.

# D   Dataset Details

## D.1   Simulation Study

We simulated datasets with four binary covariates and binary treatments and outcomes, as follows.

1. Sample covariates independently as $X_i \sim \text{Ber}(\cdot)$, where there the Bernoulli parameter for each covariate is in $[0.4, 0.5, 0.6, 0.7]$ for $i \in \{1, 2, 3, 4\}$.
2. Simulate observational data by setting a propensity score between 0 and 1 that depends on covariates $X$, given by `ps = expit ( X.dot(t-coeff) )` and sampling treatments as $T \sim \text{Ber(ps)}$. Here, we set `t-coeff` $= [0, -3, 0, 0]$ and `expit` denotes the logistic sigmoid function.
3. Assign outcomes based on treatments and covariates via a logistic model that includes non-linear interactions between treatment and covariates. Outcome $Y \sim \text{Ber ( expit (logits) )}$ where `logits = beta * T + X.dot(gamma) X.dot(delta) * T`. Here, we set `beta` $= 4$, `gamma` $= [1, -1, 1, 0]$, and `delta` $= [-2, -3, -1, -2]$.
4. To further simulate real-world selection biases, we sampled a mask $S \sim \text{Ber ( expit (sel-logits) )}$, with `sel-logits = delta-y * Y + delta-t * T + X.dot(delta-x)`, and selected only datapoints for which $S^{(i)} = 1$. Here, we set `delta-y` $= 2$, `delta-t` $= 1$, and `delta-x` $= [10, 10, 5, 1]$.

For the setting with six covariates, we take $X_5$ and $X_6$ as copies of $X_3$ and $X_4$, respectively, and similarly repeat all the corresponding coefficients that determine relationships with $T$, $Y$, and $S$. Note that our choices of coefficients here ensure that treatment effects are heterogeneous across subgroups.

## D.2   Diabetes Application

We followed the dataset curation and set-up of the Semaglutide vs. Tirzepatide experiment in Dhawan et al. (2024).

1. There are five binary covariates constructed by binning values into discrete categories:
   - `Age`: $[\leq 45, > 45]$ years,
   - `Sex`: [Male,Female],
   - `BMI` (Body Mass Index): $[\leq 28.5, > 28.5]$ kgs per meter squared,
   - `HbA1c` (Glycated Hemoglobin): $[\leq 7.5, > 7.5]$ %, and
   - `Weight`: $[\leq 220, > 220]$ lbs.
2. Possible treatments are Semaglutide $(T = 0)$ and Tirzepatide $(T = 1)$.
3. Outcome is whether the user lost 5% or more of their intital weight $(Y = 1)$ or not $(Y = 0)$.

This dataset was constructed using unstructured text data found in the PushShift collection (Baumgartner et al., 2020) of Reddit posts upto December 2022. It has been curated to contain submissions and comments that describe users' lived experiences with the treatments and outcome above. Hence, it is naturally prone to the selection biases typical of such a platform, and exemplifies a real-world use-case of the sensitivity analyses discussed in this work. Further, observational distributions were estimated by computing conditional probabilities given by a large language model (LLM), before plugging them into standard causal estimators. Specifically, we followed Dhawan et al. (2024) to sample covariates for each each report in this dataset using GPT-4 and computed the conditional distributions of treatments and outcomes given the covariates and reports using LLAMA-2-70B. This leaves the estimated distributions prone to biases of the LLMs as well.

# E    Dual Ascent for Constrained Optimization

Here we describe a standard convex optimization problem with convex constraints, present a standard dual ascent algorithm to solve it, and show that the constrained optimization problem for each of the assumptions in $(\varepsilon, \mu, p_X)$ can be written in terms of this standard problem.

$$
\begin{aligned}
\min_{\mathbf{a}} \quad & f(\mathbf{a}) \\
\text{s.t.} \quad & h_i(\mathbf{a}) \leq 0, \quad i = 1, \ldots, m, \\
& \ell_j(\mathbf{a}) = 0, \quad j = 1, \ldots, n,
\end{aligned}
$$

where $(f, h_i, \ell_j)$, are all convex functions of $\mathbf{a}$. The corresponding Lagrangian function is given by

$$
\mathcal{L}(\mathbf{a}; \mathbf{u}, \mathbf{v}) := f(\mathbf{a}) + \sum_{i=1}^{m} u_i\, h_i(\mathbf{a}) + \sum_{j=1}^{n} v_j\, \ell_j(\mathbf{a}), \qquad \mathbf{u} \geq 0,
$$

the Lagrange dual objective is $g(\mathbf{u}, \mathbf{v}) := \min_{\mathbf{a}} \mathcal{L}(\mathbf{a}; \mathbf{u}, \mathbf{v})$, and the Lagrange dual problem is

$$
\begin{aligned}
\max_{\mathbf{u}, \mathbf{v}} \quad & g(\mathbf{u}, \mathbf{v}) \\
\text{s.t.} \quad & \mathbf{u} \geq 0.
\end{aligned}
$$

Under convexity and the Slater conditions (Boyd & Vandenberghe, 2004), strong duality holds and the dual optimum equals the primal optimum. The following dual ascent algorith solves this optimization problem for differentiable $g$ and small enough learning rate $\eta$.

---

**Algorithm 1:** Dual Ascent

**Require:** Initial $\mathbf{u}^{(0)} \geq \mathbf{0}$, $\mathbf{v}^{(0)}$, learning rate $\eta$

1 **for** $k = 1, 2, \ldots$ **do**
2 $\quad$ $\mathbf{a}^{(k)} \in \arg\min_{\mathbf{a}} \mathcal{L}\big(\mathbf{a}; \mathbf{u}^{(k-1)}, \mathbf{v}^{(k-1)}\big)$
3 $\quad$ $\mathbf{u}^{(k)} \leftarrow \max\big(\mathbf{u}^{(k-1)} + \eta\, h\big(\mathbf{a}^{(k)}\big),\, \mathbf{0}\big)$
4 $\quad$ $\mathbf{v}^{(k)} \leftarrow \mathbf{v}^{(k-1)} + \eta\, \ell\big(\mathbf{a}^{(k)}\big)$
5 $\quad$ **if** *Slater's condition holds* **then**
6 $\quad\quad$ **return** $\mathbf{a}^{(k)}, \mathbf{u}^{(k)}, \mathbf{v}^{(k)}$

---

The following define convex constrained optimization problems for each of our assumption functions.

$$
\mathbf{a} = \left(\varepsilon_0(x), \frac{1}{\varepsilon_1(x)}\right),\; f(\mathbf{a}) = \|\mathbf{a} - \mathbf{1}\|_2^2,\; h_1(\mathbf{a}) = \tau(\mathbf{a}, \mu, p_X),\; h_2(\mathbf{a}) = -\mathbf{a}
$$

$$
\mathbf{a} = \mu(x),\; f(\mathbf{a}) = D_{\mathrm{KL}}(\mathbf{a}, \mu^*),\; h_1(\mathbf{a}) = \tau(\varepsilon, \mathbf{a}, p_X)
$$

$$
\mathbf{a} = p_X,\; f(\mathbf{a}) = D_{\mathrm{KL}}(\mathbf{a}, q_X),\; h_1(\mathbf{a}) = \tau(\varepsilon, \mu, \mathbf{a})
$$

Notice from eq. (2) that $\tau(\cdot)$ as given by eq. (2) is convex in each $\mathbf{a}$ defined above. Further notice that $\mathcal{L}(\cdot)$ is also convex in each $\mathbf{a}$. The optimization problems for $\mu$ and $p_X$ have additional simplex constraints, which we fold into the minimization in line 2. Hence, we can use Gradient Descent for the unconstrained minimization of $\mathcal{L}$ with respect to $\left(\varepsilon_0(x), \frac{1}{\varepsilon_1(x)}\right)$ and Entropic Mirror Descent for the simplex-constrained minimization of $\mathcal{L}$ with respect to $\mu(x)$ or $p_X$.

## F  Further Results

In our simulation study, we visualize subpopulation comparisons of sensitivity values with respect to $\varepsilon$, $\mu$, and $p_X$ in figs. 6 to 8, respectively. When comparing distributions over covariate pairs or triplets in fig. 7, the empirical BSV uncovers trends that are significantly different from not only worst-case sensitivity but also the BSV under uniform priors, highlighting the usefulness of priors that reflect real populations.

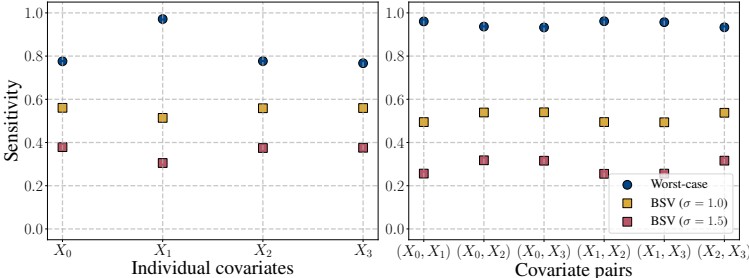

Figure 6: Worst-case sensitivity to unconfoundedness parameter $\varepsilon$ increases as number of observed confounders, and hence, dimension of $\mathcal{A}$, increases. BSV with respect to $\varepsilon(X)$ in the simulation study reveals a different trend.

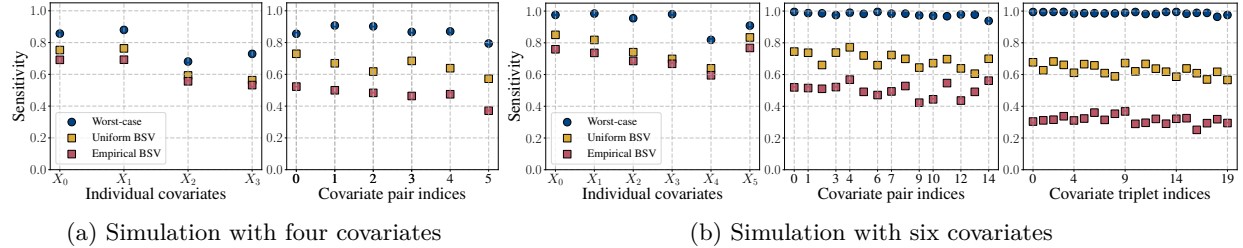

Figure 7: Worst-case sensitivity to shifts in $p_X$ over single, pairs, or triplets of covariates becomes increasingly uninformative as the dimensionality of the corresponding assumption parameter spaces increases. Bayesian sensitivity is more informative than the worst-case, with empirical priors revealing different trends than uninformative ones.

Figures 6 and 7 show increasingly high sensitivity when considering joint distributions over more covariates and can not meaningfully distinguish between different sets of covariates, especially as the dimensionality of the corresponding assumption parameter space increases. In fact, worst-case sensitivity to unconfoundedness is greater in settings with pairs of observed covariates, relative to single observed covariates, which is unintuitive and unlikely to reflect real scenarios. We attribute this limitation to the large dimension of assumption parameter spaces, where the space of possible violations grows exponentially and a worst-case analysis becomes increasingly pessimistic, regardless of the plausibility of the worst-case violation in practice. However, BSV does not suffer as much from this curse of dimensionality and better distinguishes between subpopulations even in high-dimensional parameter spaces.

High sensitivity scenarios are precisely the ones where we would like to identify subpopulations for further experimentation or data collection. The assumption on conditional outcome distributions is especially prone to exhibiting high sensitivity because shifts in $\mu$ very easily shift the ATE across the decision threshold $\delta$. Figure 8 shows worst-case sensitivity values for this parameter for different covariate and treatment values, indicating highest possible sensitivity in every case. Without incorporating any information on how likely the shifts in this parameter space are in practice, the worst-case analysis tends to be binary in nature, attributing either zero or full sensitivity to different subpopulations. This would be uninformative for practitioners trying to decide where to allocate resources for more robust ATE estimation. In contrast, BSV also reveals lower overall sensitivity for the treatment $T = 0$, whereas worst-case sensitivity values are too pessimistic to find this insight.

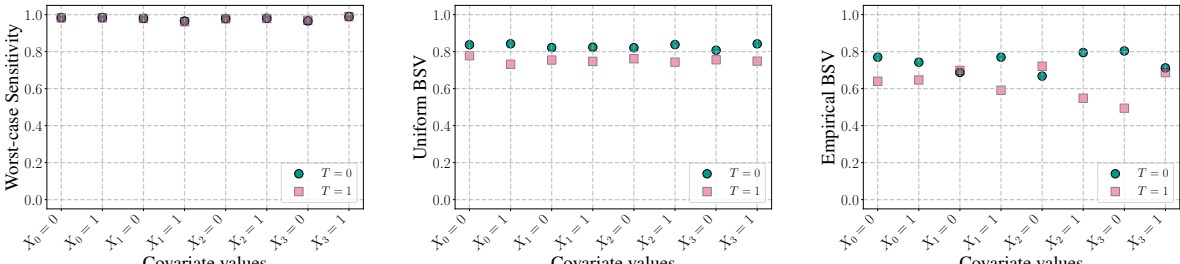

Figure 8: In the simulation study, BSV with respect to conditional outcome distributions in $\mu(X)$ attributes lower sensitivity to $T = 0$, while worst-case analyses attribute maximum possible sensitivity to all settings, making it impossible to meaningfully distinguish between subpopulations.

