# OpenReview forum: "Bayesian Sensitivity of Causal Inference Estimators under Evidence-Based Priors"
_TMLR — Accepted by TMLR_

### Review · Reviewer_yiAZ · 2025-09-16

**Summary Of Contributions:**

This paper introduces a framework to estimate the sensitivity of causal estimators to assumptions pertaining to causal data. It introduces a new criterion, Bayesian Sensitivity Value, built upon the existing s-value framework, to measure the sensitivity of an estimator based on user-defined prior distribution. The prior may be specified using domain knowledge or empirically estimated from real studies relevant to the dataset of interest. This paper applies the new framework to both a simulated and a real-world dataset and showcases the differences in the sensitivity analyses performed using BSV and the s-value (worst-case) criteria. The experiments conclude that by using BSV, one may obtain a more informative and realistic sensitivity analysis when compared to the worst-case measure.

Strengths:
- The paper is well-written with the motivation and the problem statement being very clear. The use of the real-world example and the corresponding analysis makes both the criteria and the experiments easy to follow.
- The empirical analysis of BSV for both simulated and the real-world example is thorough and well-demonstrated.
- The comparison of BSV and worst-case analysis is well-motivated and clearly articulated.

Weaknesses:
- Limited theoretical analysis of the proposed optimization (and sampling) problem: The following questions remain unanswered in the current paper: Is an exact solution always guaranteed to exist for the optimization problem? Under what conditions will this not hold, and how does that depend on the space of the assumptions $\mathcal{A}$? The intended audience may not be as familiar with optimization and sampling, and it may help to provide a simple explanation of the conditions under which you are gauranteed to reach an exact solution.
- Unclear if the benefits offered by the framework are equivalent across all types of causal estimators: the authors include the set of assumptions for the inverse propensity score weighting estimator in the Appendix, but provide limited explanation of whether the space of assumptions for the IPW estimators is equally estimable from real-world data and if the optimization problem is tractable. Similarly, a number of other estimators are commonly used (doubly robust methods, semi-parametric inference). There is no mention of whether this framework is applicable for all types of estimators and if not, what are the limitations?
- Limited applicability to other causal estimation tasks: When covariates/outcome tend to be a mixture of continuous and discrete variables, it is unclear whether the BSV and the corresponding optimization and sampling algorithms are directly applicable (or scalable).
- Limited comparison with other sensitivity analysis frameworks: Currently the new critierion is only compared to the s-value framework, however there are other sensitivity analysis methods (e.g. Tukey factorization (Franks et al., 2020)) that can also be extended to other assumptions (apart from unconfoundedness).

References
- Franks, AlexanderM, Alexander D’Amour, and Avi Feller. "Flexible sensitivity analysis for observational studies without observable implications." Journal of the American Statistical Association (2020).

**Additional Comments:**

Questions

- One of the key advantages this paper offers is an interpretable sensitivity analysis criterion that can be applied across a set of assumptions ($\epsilon, \mu, p_X$). In a way, the BSV is a way of incorporating a joint prior over all these functions. However, any sensitivity analysis that is performed in the experiments are only shown for combinations of covariates (pairs/triplets) for each function. Is there a way to extend this across functions? Can you consider a combination of $\sigma$ and $\mu$ values for the decision boundary and show this in your experiments? It would strengthen the contributions of this paper.

**Audience:**

Yes

**Audience Explanation:**

This is an important and well-motivated problem in the causal inference literature. The practical application of any proposed framework is useful to consider, and this paper does a good job of highlighting the issues with considering sensitivity analyses that are based on worst case criteria.

**Broader Impact Concerns:**

The authors do a good job of stating the broader impact concerns when applying these methods to real-world problems.

**Claims And Evidence:**

Yes

**Claims Explanation:**

The main claims made by the authors in this paper are that worst-case analysis tends to be unrealistic and any sensitivity analysis framework considering only the covariate distribution may be incomplete for a causal estimation task. The authors support this claim by showing that, for an outcome imputation estimator, there are a number of additional assumptions that tend to be made in practice (unconfoundedness and well-specified outcome models) which can be incorporated into the sensitivity analysis frameworks.

However, I would recommend that the authors be explicit about the set of causal estimators to which their framework applies as they state that their framework is a general sensitivity analysis framework for causal estimators. From the current version of the paper, it is not clear whether the space of assumptions for all estimators leads to a formulation of the constrained optimization problem which can be easily solved. (See details in the Requested Changes).

**Requested Changes:**

Major Revisions (Prior to acceptance)

- Comprehensive comparison to other sensitivity analysis frameworks: In the list of the contributions, the authors state "We unify sensitivity analysis frameworks with respect to different types of assumptions in a general framework that allows practitioners to ask targeted questions about the sensitivity of a study." This seems more general than the scope of the current paper. There are other ways to do sensitivity analysis which treats the causal inference task as a missing data problem (e.g., Franks et al., 2020). Such methods provide a confidence interval for the causal effect based on the possible bounds of the parameters. These methods may also be extended to other assumptions (and not just unconfoundedness) and can also be translated to the decision-making setting. Therefore, I believe that the current framework, while extending the s-value framework to assumptions pertaining to causal estimators is still not unifying all possible ways of conducting sensitivity analysis. I would like to at least see a comparison of the advantages and disadvantages of the BSV criteria when compared to other sensitivity analysis methods, and if feasible (using the same set of assumptions) an empirical comparison as well.
- Rigorous theoretical analysis of the constrained optimization problem: Is it guaranteed that we will reach the optima in the current setting? Under what conditions will that not hold true? I would like to see an explanation in the main text (and additional theorems in the appendix for any proofs, if required).
- Clarification on the scope of the causal estimation task: While the paper explicitly focuses on the case of discrete variables for all (covariates/treatment/outcome), there is no mention of whether this framework is generalizable to other cases. Is the recommendation that all variables have to be discretized appropriately for this framework to be applied? If so, this has to be explicitly stated in Section 2.1.
- Clarification on the types of causal estimators for which such a framework is applicable: There are a number of other causal estimators which are in use in practice today (e.g. BART, double machine learning). The paper focuses on two estimators (outcome imputation and IPW), but does not clarify whether it is generalizable, and what are the barriers to generalization, if any. This would be required for this framework to be a "general framework for sensitivity analysis". Specifically, include whether the space of assumptions for other estimators can be broken down to a discrete, bounded space for the optimization problem to be solvable.

Minor Revisions (Good to include)

The following parts can to be rewritten/improved for clarity
- Introductory graphic. Figure 1 is difficult to follow along with the Introduction (as the meaning of $\pi$ etc is not explained). I would improve the explanation in the caption or use a simpler graphic to explain the concept.
- The current writing assumes that the users have an understanding of the s-value framework in the beginning, which may not be true. I would summarize the framework in a couple of sentences in the introduction.
- Most of the introduction focuses on the usefulness of deriving empirical priors from real-world data, but this is not possible for all assumptions ($\epsilon$). I would state that upfront in Section 2.1 (instead of in the experiments section).
- Experiments: It is not immediately clear what the covariate pair indices are in Figure 5.

Provide intuition for some of the observations: In the experiments, you mention that BSV is able to provide trends for $\mathcal{A}$ even in high dimensions. It would be good to include some intuition for why this is the case (with respect to either the sampling or constrained algorithms).

---

> ### Author Response · Authors · 2025-11-05
> **Author Response (Part 1)**
>
> Thank you for your positive review and detailed feedback! We address each question and suggestion below. Corresponding updates to our paper are highlighted in blue in the revised draft.
>
> > Comprehensive comparison to other sensitivity analysis frameworks: In the list of the contributions, the authors state "We unify sensitivity analysis frameworks with respect to different types of assumptions in a general framework that allows practitioners to ask targeted questions about the sensitivity of a study." This seems more general than the scope of the current paper. There are other ways to do sensitivity analysis which treats the causal inference task as a missing data problem (e.g., Franks et al., 2020). Such methods provide a confidence interval for the causal effect based on the possible bounds of the parameters. These methods may also be extended to other assumptions (and not just unconfoundedness) and can also be translated to the decision-making setting. Therefore, I believe that the current framework, while extending the s-value framework to assumptions pertaining to causal estimators is still not unifying all possible ways of conducting sensitivity analysis. I would like to at least see a comparison of the advantages and disadvantages of the BSV criteria when compared to other sensitivity analysis methods, and if feasible (using the same set of assumptions) an empirical comparison as well.
>
> Thank you for pointing out this related work! First, we would like to clarify that we do not want to claim a unification of all sensitivity analyses possible, but instead a framework to unify and treat different assumptions required by a given estimator. In fact, Franks et al., 2020 uses some distributional assumptions in different examples, which do not fit directly into our framework. We have edited our statement of contributions to communicate this better.
> The sensitivity analysis of Franks et al., 2020 is with respect to parameters $(\gamma_0, \gamma_1)$ which represent unconfoundedness (when they are set to 1) and can be interpreted as the differences of expected conditional potential outcomes instead of the ratios $(\varepsilon_0, \varepsilon_1)$ proposed by Lu and Ding, 2023 [1]. One key advantage of our framework over these two prior works is its ability to maintain the dependence of these parameters on covariates $X$. We have included this discussion in Section 2.3.
>
> > Rigorous theoretical analysis of the constrained optimization problem: Is it guaranteed that we will reach the optima in the current setting? Under what conditions will that not hold true? I would like to see an explanation in the main text (and additional theorems in the appendix for any proofs, if required).
>
> The standard constrained optimization algorithm presented in the paper can be extended and applied to any causal estimand that can be written as a convex function of its assumption $a$, divergence $D$ that is convex in $a$, and convex assumption space $\mathcal{A}$. We have clarified this in Section 4, along with detailed discussion of the algorithm in Appendix E.
>
> > Clarification on the scope of the causal estimation task: While the paper explicitly focuses on the case of discrete variables for all (covariates/treatment/outcome), there is no mention of whether this framework is generalizable to other cases. Is the recommendation that all variables have to be discretized appropriately for this framework to be applied? If so, this has to be explicitly stated in Section 2.1.
>
> You are right in that we present practical algorithms with a focus on the setting with binary treatments and outcomes and discrete covariates, leaving extensions to continuous-valued variables to future work, which is now made explicit in Section 2.1.
>
> > Clarification on the types of causal estimators for which such a framework is applicable: There are a number of other causal estimators which are in use in practice today (e.g. BART, double machine learning). The paper focuses on two estimators (outcome imputation and IPW), but does not clarify whether it is generalizable, and what are the barriers to generalization, if any. This would be required for this framework to be a "general framework for sensitivity analysis". Specifically, include whether the space of assumptions for other estimators can be broken down to a discrete, bounded space for the optimization problem to be solvable.
>
> Our framework is directly applicable to causal estimators as long as we can write the causal estimand, e.g. ATE, as a function of the assumption. Our optimization algorithms rely on this function being convex in the assumption and on the convexity of the divergence and assumption space, but approximate solutions may be possible with algorithms that extend to non-convex settings. We have made this explicit at the start of section 4 and modified the language through the paper (including the contributions) to clarify the applicability of our framework.

---

> > ### Author Response · Authors · 2025-11-05
> > **Author Response (Part 2)**
> >
> > > Introductory graphic. Figure 1 is difficult to follow along with the Introduction (as the meaning of $\pi$ etc is not explained). I would improve the explanation in the caption or use a simpler graphic to explain the concept.
> >
> > Thank you for pointing this out; we have now defined $\pi$ and added details in the caption.
> >
> > > The current writing assumes that the users have an understanding of the s-value framework in the beginning, which may not be true. I would summarize the framework in a couple of sentences in the introduction.
> >
> > We have added a short description of the $s$-value in the third paragraph of the Introduction. We hope this also helps motivate the demonstration in Figure 1.
> >
> > > Most of the introduction focuses on the usefulness of deriving empirical priors from real-world data, but this is not possible for all assumptions ($\varepsilon, \mu, p_X$). I would state that upfront in Section 2.1 (instead of in the experiments section).
> >
> > We have now included a statement of this distinction in Section 2.1.
> >
> > > Experiments: It is not immediately clear what the covariate pair indices are in Figure 5.
> >
> > In order to avoid clutter in the figure itself, we have added a description explaining the “Covariate pair indices” in Section 6.3, where this figure is referenced.
> >
> > > One of the key advantages this paper offers is an interpretable sensitivity analysis criterion that can be applied across a set of assumptions ($\varepsilon, \mu, p_X$). In a way, the BSV is a way of incorporating a joint prior over all these functions. However, any sensitivity analysis that is performed in the experiments are only shown for combinations of covariates (pairs/triplets) for each function. Is there a way to extend this across functions? Can you consider a combination of $\sigma$ and $\mu$ values for the decision boundary and show this in your experiments? It would strengthen the contributions of this paper.
> >
> > This is definitely an interesting question! Currently, we opt to treat each assumption separately for ease of interpretation. Further, when possible assumption values exist in different spaces with different divergences, it is not immediately obvious how to combine these spaces and divergences in order to combine the assumptions in a single criterion. We agree that the BSV enables us to readily incorporate a joint prior over all assumptions, and extending it to handle multiple assumption spaces and divergences while maintaining interpretability would be interesting future work.
> >
> > [1] Sizhu Lu and Peng Ding. Flexible sensitivity analysis for causal inference in observational studies subject to unmeasured confounding. arXiv preprint arXiv:2305.17643, 2023.
> >
> > We really appreciate the detailed feedback and believe it has greatly improved the presentation in our paper. Please let us know if there are any other questions or concerns we can address.

---

> > > ### Comment · Reviewer_yiAZ · 2025-11-09
> > >
> > > Thank you for the detailed explanation for each point that was raised in the review. The revised text and additional experiments improve the clarity of the idea as well as the exact contributions of this work.

---

### Review · Reviewer_BP7m · 2025-10-06

**Summary Of Contributions:**

This paper extends the recent s-value framework to a broader class of assumptions and introduce a new Bayesian sensitivity criterion: the Bayesian Sensitivity Value (BSV). BSV computes the expected sensitivity under prior distributions from real-world data. They provide algorithms for computing both worst-case and Bayesian sen

**Additional Comments:**

The paper mentions that observational distributions in the diabetes treatment case study are extracted using a large language model from Reddit data (Pushshift dataset). However, this procedure is not clearly presented. It is suggested to discuss this, such as the specific role and configuration of the LLM in the pipeline and evidence of reliability.

**Audience:**

Yes

**Audience Explanation:**

The causal inference community will be interested in this paper.

**Claims And Evidence:**

Yes

**Claims Explanation:**

1. This paper is motivated by the limitations of existing sensitivity analysis frameworks, which typically focus on worst-case assumption violations. Such approaches can be overly pessimistic and may not reflect realistic scenarios. The motivation is effectively illustrated in Figure 1 and Figure 2.
2. The authors extend the s-value framework beyond covariate distribution shifts by incorporating evidence-based priors. The proposed BSV is mathematically formulated in Equation 7, which generalizes the original s-value formulation to a broader class of assumptions.
3. Two algorithmic approaches are proposed to compute sensitivity values under the BSV framework: constrained optimization for worst-case sensitivity and Monte Carlo sampling for Bayesian sensitivity. These practical algorithms are detailed in Section 4.
4. The proposed framework is empirically validated through both simulation studies and a real-world diabetes treatment case study. As shown in Section 6, BSV consistently provides more nuanced and realistic sensitivity assessments compared to worst-case analyses.

**Requested Changes:**

1. The paper demonstrates BSV primarily under empirical or data-driven priors, but does not evaluate how BSV behaves under prior misspecification, which is a realistic concern in applied settings. It is recommended to study the BSV under a range of prior distributions and the results under prior misspecification.
2. It is interesting to discuss whether the proposed method is applicable to other unidentifiable causal inference conditions.

---

> ### Author Response · Authors · 2025-11-05
> **Author Response**
>
> Thank you for your encouraging review of our work and helpful suggestions to improve it! Below we address the requested changes and corresponding updates to the paper, highlighted in the revised draft in blue.
>
> > The paper demonstrates BSV primarily under empirical or data-driven priors, but does not evaluate how BSV behaves under prior misspecification, which is a realistic concern in applied settings. It is recommended to study the BSV under a range of prior distributions and the results under prior misspecification.
>
> To demonstrate behaviour of the BSV under different priors, we include a data-driven as well as uninformative (uniform) prior in Figure 3. Figure 5 also shows the Uniform BSV for different assumptions in the Diabetes application. At a high level, we would like to reiterate that the BSV is prior-dependent. In fact, if we choose the prior that puts all probability mass on the worst-case value of the assumption, we would recover the worst-case sensitivity value. We have further clarified this and highlighted the reliance of the BSV on priors in our Limitations section.
>
> > It is interesting to discuss whether the proposed method is applicable to other unidentifiable causal inference conditions.
>
> Our method is directly applicable to causal inference assumptions as long as we can write the causal estimand, e.g. ATE, as a function of the assumption. We show this for the outcome imputation estimator in the main paper and include inverse propensity score weighting as another example in Appendix C. Our optimization algorithms rely on this function being convex in the assumption, but approximate solutions may be possible with algorithms that extend to non-convex settings. For the BSV, in particular, we require access to a prior over the assumption variable, which may be constructed in a data-driven way if possible, or else may be the uninformative uniform prior. We have now clarified this at the start of section 4.
>
> > The paper mentions that observational distributions in the diabetes treatment case study are extracted using a large language model from Reddit data (Pushshift dataset). However, this procedure is not clearly presented. It is suggested to discuss this, such as the specific role and configuration of the LLM in the pipeline and evidence of reliability.
>
> Thank you for pointing this out! We have now added more details about the experimental configuration of the Diabetes application in Appendix D.2, following the set up of the previous work that proposed this dataset.
>
>
> Thank you for these suggestions to improve our paper. Please let us know if there is anything else we can clarify.

---

> > ### Comment · Reviewer_BP7m · 2025-12-02
> > **Reply to the author response**
> >
> > Thanks for the clarification and response. I have no further questions.

---

### Review · Reviewer_UfqP · 2025-10-27

**Summary Of Contributions:**

This paper recasts three common causal assumptions on unmeasured confounding, outcome models, and covariate distribution into a single assumption-space framework. The authors show that the worst-case sensitivity is a special case that relies on unrealistic changes in the data-generating process. They proposed a Bayesian sensitivity value that averages $\exp(-D)$ over assumption draws from a prior restricted to the decision-reversing region, with an empirical Bayes version using external datasets such as NHANES, UCI, and subgroup analyses as priors. They provide optimization and sampling routines and a diabetes case study to argue BSV is more informative than worst-case, especially as the dimensionality of the assumption space grows.

Key strengths:
Nicely argued motivation that worst-case optima can be implausible under empirical knowledge.
A clear, unified notation of the sensitivity parameters across different models.
An optimization toolkit.

Weakness:
The BSV as a scalar is not very well calibrated, prior‑sensitive, and hard to compare across assumptions.
The empirical case mixes heterogeneous evidence with minimal transport diagnostics.

**Audience:**

Yes

**Audience Explanation:**

I think so. The work touches a live pain point that worst-case sensitivity often leads to too conservative conclusions, and it offers a general, implementable alternative with appealing visuals. Hope the unifying formulation and the idea of empirical priors for sensitivity are attractive to causal ML and applied researchers.

**Broader Impact Concerns:**

As mentioned above, I would be concerned if the prior for $\mu$ is specified only for one treatment arm, since this could inadvertently favor the arm with a richer or better informed prior.

Also the BSV is conditional on $\tau(A)\leq \delta$, stakeholders may misinterpret it as reflecting the overall robustness of the analysis while overlooking how often such decision reversals actually occur.

**Claims And Evidence:**

Yes

**Claims Explanation:**

The claims are mostly accurate and convincing. For example, the paper shows that the worst-case optima can be implausible and that the worst-case sensitivity tends to saturate in their setups. They also generalize s-value notation cleanly, and the "expectation <= supremum" observation correctly implies BSV will be no more pessimistic.

**Requested Changes:**

I think the paper can be significantly strengthened by addressing the following substantive issues:

1. The paper proposes to pool the three assumptions ($\epsilon, \mu, p_X$) and compute the generalized s-value jointly. However, these components differ fundamentally: $\mu$ and $p_X$ can, at least in principle, be tested and/or validated using the observed data, while $\epsilon$ is a causal identification assumption that is inherently untestable at all from the observed data. Treating all three within a single metric may obscure this asymmetry and make interpretation difficult. It would strengthen the paper to explicitly discuss this distinction, both conceptually and methodologically, and to consider whether it calls for different treatment (e.g., distinct priors, divergences, or sensitivity scales) for these assumption types.

2. The empirical results illustrate that BSV can yield more discriminative values than worst-case sensitivity in the presented settings. However, the evidence of reliability or superiority beyond carefully chosen priors and discretizations is limited. The current conclusions seem heavily dependent on specific prior shapes (e.g., truncated Gaussians for $\epsilon$) and binarized covariates. Including prior-sensitivity or stability analyses would greatly strengthen the methodological credibility and help clarify when BSV offers genuine robustness rather than prior-driven variation.

3. Relevantly, the metric's semantics are weak, the role of priors is under-validated, and the diabetes case leans on LLM-extracted observational quantities from Reddit plus priors from unrelated cohorts, without transport diagnostics. I would suggest reporting $Pr(\tau(A)\leq \delta)$ and BSV jointly, and add uncertainty and acceptance rates.

---

> ### Author Response · Authors · 2025-11-05
> **Author Response**
>
> Thank you for the positive review and thoughtful suggestions to improve our paper! Below, we address each of the requested changes and have highlighted corresponding updates to our draft in blue.
>
> >  The paper proposes to pool the three assumptions ($\varepsilon, \mu, p_X$) and compute the generalized s-value jointly. However, these components differ fundamentally: $\mu$ and $p_X$ can, at least in principle, be tested and/or validated using the observed data, while $\varepsilon$ is a causal identification assumption that is inherently untestable at all from the observed data. Treating all three within a single metric may obscure this asymmetry and make interpretation difficult. It would strengthen the paper to explicitly discuss this distinction, both conceptually and methodologically, and to consider whether it calls for different treatment (e.g., distinct priors, divergences, or sensitivity scales) for these assumption types.
>
> Thanks for raising this point! We would like to clarify that we do not, in fact, compute the generalized $s$-value or the BSV jointly for all three assumptions. As you pointed out, this would make interpretation difficult. We have clarified the writing at the start of Section 2 to present this better. We do treat each assumption separately, with its own space of possible values $\mathcal{A}$ and divergence $D$, as detailed in Table 1. We also agree that $\mu$ and $p_X$ do differ fundamentally from $\varepsilon$ in terms of observability, and have now explicitly stated this in Section 2.1.
>
> > The empirical results illustrate that BSV can yield more discriminative values than worst-case sensitivity in the presented settings. However, the evidence of reliability or superiority beyond carefully chosen priors and discretizations is limited. The current conclusions seem heavily dependent on specific prior shapes (e.g., truncated Gaussians for ) and binarized covariates. Including prior-sensitivity or stability analyses would greatly strengthen the methodological credibility and help clarify when BSV offers genuine robustness rather than prior-driven variation.
>
> To demonstrate trends in the BSV under different priors, we include a data-driven as well as uninformative (uniform) prior in Figure 3. At a high level, it is definitely true that the BSV is prior-dependent and reflects its variations. In fact, if we choose the prior that puts all probability mass on the worst-case value of the assumption, we would recover the worst-case sensitivity value. We have further clarified this and highlighted the reliance of the BSV on its priors in our Limitations section.
>
> > Relevantly, the metric's semantics are weak, the role of priors is under-validated, and the diabetes case leans on LLM-extracted observational quantities from Reddit plus priors from unrelated cohorts, without transport diagnostics. I would suggest reporting $Pr(\tau(A) \leq \delta)$ and BSV jointly, and add uncertainty and acceptance rates.
>
> Thank you for this suggestion! Reporting $P(\tau(A) \leq \delta)$ along with the BSV is definitely more informative since the former accounts for how likely decision reversals are and the latter provides information about the average “size” (via divergence $D$) of the assumption violation that reverses the decision. We report the BSV, acceptance rates, and $P(\tau(A) \leq \delta)$ for the Diabetes case study in Table 2.
> Please let us know if there are any other diagnostics or uncertainty measures that would improve the paper.
>
> > As mentioned above, I would be concerned if the prior for $\mu$ is specified only for one treatment arm, since this could inadvertently favor the arm with a richer or better informed prior.
>
> We agree with this concern. As discussed in the Broader Impacts Statement, our framework should not be used if it is not reasonable or possible to construct a prior for some assumption variable. In the case of $\mu$, we would recommend using BSV only for the treatment arm $t$ for which we are able to construct a prior, i.e. take $A$ as just $\mu_t$.
>
> > Also the BSV is conditional on $\tau(A) \leq \delta$, stakeholders may misinterpret it as reflecting the overall robustness of the analysis while overlooking how often such decision reversals actually occur.
>
> Thank you; this is a great point. As discussed above, we now report $P(\tau(A) \leq \delta)$ in addition to the BSV.
>
>
> We appreciate all the suggestions and believe that they improve the paper. Please let us know if there are any further questions we can address.

---

### Author Response · Authors · 2025-11-05
**Summary of Updates**

We thank the reviewers for their time and the effort they took to provide insightful feedback! Reviewer feedback has helped us identify a few ways to improve the paper. Based on the reviewers’ suggestions, we would like to include the following edits, which are highlighted in blue in our updated draft. We expand on these points in more detailed reviewer responses in the comments below.

1. **Clarification on the scope of our framework**: We clarify in section 4 of the paper that our framework is directly applicable to causal estimators as long as we can write the causal estimand, e.g. ATE, as a function of the assumption of interest. Our optimization algorithms rely on this function being convex in the assumption and on the convexity of the divergence and assumption space.

2. **Constrained optimization algorithms**: We include the standard dual ascent algorithm in Appendix E, together with details on how to write our constrained optimization problems in terms of the standard problem.

3. **Role of priors**: We clarify through the work and in our Limitations section that the BSV is prior-dependent. In our experiments, we include results under the uninformative (uniform) prior, as shown in Figures 3 and 5 and Table 2.

4. **Acceptance rate and probability of decision reversal**: We include acceptance rates and the probability of decision reversal, $P(\tau(A) \leq \delta)$, along with the BSV, for our real-world Diabetes experiment, in Table 2 of the revised draft. We believe this improves the sensitivity analysis by providing information on how likely causal decision reversals are along with the average extent of assumption violations that reverse the decision.

---

### Decision · Action_Editor_qAnr · 2025-12-21

**Recommendation:** Accept as is

**Audience:**

Yes

**Audience Explanation:**

Yes, it appeals to sensitivity analysis more broadly. This would appeal to all ML researchers working in causal inference.

**Claims And Evidence:**

Yes

**Claims Explanation:**

The main claim is that sensitivity analysis, such as that based on s-value, is worst-case and its applicability can be improved by adding a Bayesian prior.
The claim is supported by the analysis of s-value in the paper and subsequently through empirical experiments. The paper makes a convincing case on how the current methods such as s-value correspond to a worst-case analysis and provide two algorithmic approaches to compute the Bayesian sensitivity value.
In addition, the paper shows how the prior can be derived using observational data, which is useful for practical implementation.